# VALUE FROM OBSERVATIONS: TOWARDS LARGE-SCALE IMITATION LEARNING VIA SELF-IMPROVEMENT

## ABSTRACT

Imitation Learning from Observation (IfO) offers a powerful way to learn behaviors from large-scale, mixed-quality data. Unlike behavior cloning or offline reinforcement learning, IfO leverages action-free demonstrations and circumvents the need for costly action-labeled demonstrations or carefully crafted reward functions. However, current research focuses on idealized scenarios with tailored, bimodal-quality data distributions. This paper introduces a novel algorithm to learn from datasets with varying quality, moving closer to a paradigm in which imitation learning can be performed iteratively via self-improvement. Our method extends RL-based imitation learning to action-free demonstrations, using a value function to transfer information between expert and non-expert data. Through comprehensive evaluation, we delineate the relation between different data distributions and the applicability of algorithms and highlight the limitations of established methods. Our findings provide valuable insights for developing more robust and practical IfO techniques on a path to scalable behaviour learning.

## 1 INTRODUCTION

Bringing the cognitive capabilities of large vision and language models to embodied systems is at the forefront of many researchers' attention. Given the nature of the underlying data, robotics has been a popular target domain with a variety of deployed approaches, including prompting based algorithms (Ahn et al., 2022; Jiang et al., 2023; Kwon et al., 2024; Di Palo et al., 2023) and behavior cloning (BC) (Brohan et al., 2022; Bousmalis et al., 2024). We are motivated by a practical future where agents are trained on a large-scale dataset of language and video without requiring explicit action annotations or targeted task prompts. If required, the agent should further autonomously collect data to address knowledge gaps. Enabling learning from such data could overcome two key hurdles to scaling imitation learning: the static limitations of prompt engineering that may hinder generalization and the high cost of large-scale action-labeled demonstrations.

As a first step towards this setting, this paper studies the problem of training agents assuming access to two kinds of data: *expert* demonstrations without action labels, and *background* datasets which include actions but do not necessarily solve any task. So while the agent has access to some action data, it has not directly experienced the task being solved (it may only see solutions within the action-free *expert* data) and does not observe any external rewards. This setup can be directly applied to settings where action labels are difficult to obtain or which would benefit from cross-embodiment transfer, e.g. autonomous driving datasets (CAR) or data collected with the UMI-gripper (Chi et al., 2024). Expanding the *expert* data in our problem setting to include in-the-wild observations of humans solving tasks would allow scaling to cheaply available large-scale data.

Any applicable method has to address two key challenges: a) collect useful data for imitating the expert behaviour and b) learn a robust imitation policy from that data. Collecting data and inferring policies is at the heart of reinforcement learning (RL). Driven by a reward function, RL agents collect data and improve a policy based on the collected, possibly non-expert, experience. The reward serves as key signal for improvement and represents the main judge of *good* and *bad*. However, engineering a reward model can be time intensive (Tirumala et al., 2024; OpenAI et al., 2019; Lee et al., 2020) and cannot capture the breadth of possible contexts and tasks encountered at scale. We are thus

interested in algorithms that acquire a notion of *good* and *bad* directly from data. This brings us to imitation learning, in particular Imitation learning from Observations (IfO) which addresses the problem of imitating expert behavior without requiring action annotations (Torabi et al., 2018a;b; Liu et al., 2018). To date, IfO remains an open research problem (Sikchi et al., 2024) and current methods do not match the maturity of related methods in RL, which have been shown to scale to large datasets and models (Springenberg et al., 2024; Chebotar et al., 2021). In contrast, IfO methods have been benchmarked only in small, single-domain settings with ad-hoc choices for data generation (Ma et al., 2022; Zolna et al., 2020; Sikchi et al., 2024). We here suggest a concrete benchmark that moves towards data compositions that better reflect the above large-scale vision.

Concretely, previous work has mainly been focused on offline training on mixed data where a significant amount of expert data was diluted in non-expert data (Ma et al., 2022; Zolna et al., 2020; Sikchi et al., 2024). Instead, we extend previous datasets (Fu et al., 2021) and collect data with a variety of policies of different quality and examine how much performance can be improved across this spectrum. We argue that this setup more accurately captures the scenario where an agent collects its own data for improvement and we underline this with a self-improvement experiment where an agent uses self-collected data to improve.

In addition to forming a new benchmark, we propose a simple, offline IfO method. Our algorithm adapts either SQIL (Reddy et al., 2020) or ORIL (Zolna et al., 2020), two RL-based imitation learning algorithms, to the action free setting. For the SQIL based variant, we simply assign a reward of 1 to *expert* data and a reward of 0 to *background* data before applying a value-function based offline RL method akin to AWR (Wang et al., 2016; Peng et al., 2019). The use of a value function instead of state-action value (aka a Q function) overcomes the lack of expert action annotations. In the second variant, we replace the 0-1 rewards with estimates from a learned discriminator as in Ho & Ermon (2016); Zolna et al. (2020). In both variants, we combine a simple reward function with a learned value-function which transfers expert knowledge from the unlabeled expert data onto the action labeled *background* data. We name our approach Value learning from Observations (VfO).

In summary, we suggest the combination of IfO and iterative self-improvement in order to approach large scale behavior learning from a novel angle and under realistic, scalable data collection assumptions. To this end we introduce a new offline benchmark that is more representative of said setting. We further propose a novel algorithm (VfO), in two variants, that adapts offline RL mechanisms to imitation learning from observations. With a broad set of experiments, we confirm the representative power of our benchmark, underline the competitiveness of our algorithm, and show initial positive results of IfO in conjunction with iterative self-improvement.

## 2 RELATED WORK

The classical, straightforward approach to imitation learning is behaviour cloning (BC, Osa et al. (2018)), i.e., maximising the likelihood of actions in the dataset. However, this approach requires large numbers of optimal demonstrations. For this reason, a variety of methods have been developed to additionally benefit from suboptimal and other data sources by for instance extrapolating rewards from observations (Brown et al., 2019; Chen et al., 2021), imitation via IRL (Davchev et al., 2021a), or even by using videos from generative models (Bharadhwaj et al., 2024).

In online imitation learning, self-generated agent data represents the best data distribution to learn how to refine agent behaviour (Ross et al., 2011; Swamy et al., 2022; Lavington et al., 2022). Different methods have been proposed to apply the reinforcement learning formalism to address an imitation problem - from classical and deep maximum entropy inverse RL (Ziebart et al., 2008; Wulfmeier et al., 2015; Barnes et al., 2024) to computationally more efficient adversarial imitation learning (Ho & Ermon, 2016; Fu et al., 2017; Wulfmeier et al., 2017; Kostrikov et al., 2019a). When treating imitation as matching of agent visited transitions or divergence minimisation, further divergences have been explored (Ke et al., 2021; Ghasemipour et al., 2020). Transitioning from adversarial learning to more stable value function optimisation, SQIL removes the intermediary classifier from GAIL and instead uses a binary reward (Reddy et al., 2020). IQlearn (Garg et al., 2021) and ValueDICE (Kostrikov et al., 2020) enable transitioning from explicitly defined to implicitly learned rewards. Other non adversarial algorithms include PPIL (Viano et al., 2022), PWIL (Dadashi et al., 2021), and CSIL (Watson et al., 2024). However, online imitation learning can be costly in domains like robotics, is sometimes not even possible and doesn't benefit from existing data sources.

When online data generation is impractical, suboptimal offline datasets can provide an alternative. Practical and scalable algorithms can be derived when using either discriminator (Zolna et al., 2020) or optimal transport (Luo et al., 2023) based rewards together with offline RL. Value (Kim et al., 2022) and model-based approaches (Chang et al., 2021) expand the toolkit. IQLearn can further be shown to be equivalent to BC with dynamics-aware regularisation term (Wulfmeier et al., 2024).

The online and offline settings described above require access to high-quality demonstrations with action annotations, often only attainable via complex tele-operation settings in robotics. The extension towards action-free demonstrations opens considerable scope and has been the target of further methods. A separately trained inverse dynamics model can be applied to label action-free data, enabling behaviour cloning (Radosavovic et al., 2021; Torabi et al., 2018b). Learning rewards provides a path to instead relabel sub-optimal data for RL style optimisation (Eysenbach et al., 2021; Davchev et al., 2021b). Here, adversarial approaches present a common mechanism to learn rewards (Ho & Ermon, 2016). These can be adapted by controlling the discriminator input space, often benefiting from further regularisation (Zhu et al., 2020b; Liu et al., 2020). Value function based imitation methods enabled by inverse Bellman updates and dual formulations of the problem like SMODICE (Ma et al., 2022) and DILO (Sikchi et al., 2024), or variational formulations (Kostrikov et al., 2019b; Garg et al., 2021), bypass the often hard to optimize adversarial objectives and are related to our approach. While mathematically appealing, these methods can still be brittle and harder to scale to the real-world directly from raw observations (Al-Hafez et al., 2023; Watson et al., 2024). Instead, we base our value-based algorithm on a simple RL backbone which draws on decades of experience. We compare performance and show competitiveness against various baselines (including SMODICE and DILO) on a large set of experiments on different domains.

## 3 METHOD

### 3.1 OFFLINE IMITATION LEARNING FROM OBSERVATIONS

We consider learning in a dynamical system modelled as Markov decision process with states $s \in S$, actions $a \in A$, and dynamics $p(s_{t+1}|s_t, a_t)$. In order to learn useful behaviour, the agent has access to two sources of information: a dataset of *expert* state trajectories $\tau_E = (s_1, \ldots, s_T) \in D_E$ without actions and a dataset of state-action trajectories from its own embodiment but of unspecified origin and quality $\tau_B = (s_1, a_1, \ldots, a_{T-1}, s_T) \in D_B$. We will refer to the latter as *background* dataset. The agent's goal is to obtain a policy $\pi(a|s)$ that imitates the behaviour underlying the expert trajectories. To support scalability, we limit the use of further information (e.g. rewards, domain knowledge) and thus do also not expect the agent to outperform expert performance.

Given the lack of *expert* actions, the agent has to be able to leverage the *background* dataset to understand the dynamics, i.e., the relationship between actions and states. However, similar to prior work on inverse RL (Abbeel & Ng, 2004; Ziebart et al., 2008) as well as for the related problem of offline RL (Schweighofer et al., 2022; Hong et al., 2023), the quality and distribution of the *background* data plays an important role on the achievable performance. In previous work, different sources have been employed, such as agent replay data, a mixture of expert and non-expert data, or data collected with a suboptimal policy. Given that we are interested in an agent that can start from few assumptions and that should be able to leverage the data it collects, we focus on the suboptimal policy case. In order to generate a corresponding benchmark we suggest to train multiple policies using BC but vary the number of demonstrations provided. We then run these policies to collect multiple datasets of varying quality (see Section 4.1).

### 3.2 VFO: VALUE FROM OBSERVATION

We introduce a simple IfO method that can effectively learn from observations in the self-improvement setting at hand. For this purpose, we consider two variants of a value-function based approach that learns a state-value function from observations alone. In the first, we assign binary rewards (i.e., 1 for *expert* or 0 for *background*) to the data – thus adapting SQIL (Reddy et al., 2020) to our setting. In the second, we use a learned discriminator that performs a soft *expert* / *background* assignment of each state – thus adapting ORIL (Zolna et al., 2020) to our setting. Other imitation learning-based rewards such as Luo et al. (2023) could also be employed, but note that employing rewards that rely on prior knowledge such as when derived from goal states may impact generality.

---

**Algorithm 1** Value from Observation (VfO)

---

**Require:** Expert dataset $D_E$, background dataset $D_B$, mixture parameter $\alpha$, temperature $\lambda$, discount $\gamma$, initial policy $\pi_0$, initial value $v_0$. **Optional:** discriminator $d(s) : S \mapsto [0, 1]$

**for** $k \leftarrow 1$ to $K$ iterations **do**

$$r(s', z) = \begin{cases} d(s') & \text{if discriminator provided} \\ \mathbf{1}_E(z) & \text{otherwise} \end{cases} \qquad \triangleright \mathbf{1}_E \text{ is the expert indicator function.}$$

$L_k^v \leftarrow E_{(s,s',z)\sim(1-\alpha)D_E+\alpha D_B} \left(\gamma v_{k-1}(s') + r(s', z) - v_{k-1}(s)\right)^2$

$L_k^\pi \leftarrow -E_{(s,s',a,z)\in D_B} \exp\left((\gamma v_{k-1}(s') + r(s', z) - v_{k-1}(s))/\lambda\right) \log(\pi_{k-1}(a|s))$

$v_k \leftarrow \text{AdamUpdate}(v_{k-1}, L_k^v), \pi_k \leftarrow \text{AdamUpdate}(\pi_{k-1}, L_k^\pi)$

**end for**

---

As mentioned above, we resort to learning a state-value function for transferring knowledge from the *expert* data without action labels to the *background* data. We note that a state-action Q-value based offline RL approach cannot be applied in our setting due to a lack of signal on the *background* data: all transitions are labeled with a zero reward in the binary setting or potentially very small rewards in the learned discriminator setting. In contrast, if we apply an approach based on the state-value function $v$, policy evaluation is possible without knowing the action and can thus leverage a mixture of *expert* and *background* data.

First, we define a virtual policy $\bar{\pi}$ which mixes the *expert* and *background* data-generating processes at each transition:

$$\bar{\pi}(a|s) = p(z = E|s, \alpha)\pi_E(a|s) + p(z = B|s, \alpha)\pi_B(a|s) \tag{1}$$

with mixture coefficient $\alpha$ and where $z$ denotes the latent indicating the origin of the data, either *expert* $E$ or *background* $B$. This policy is equivalent to deciding at the beginning of an episode whether to follow the implicit expert or background policy. Note that the probability of using $\pi_E$ or $\pi_B$ is state-dependent, and will depend on the likelihood of reaching $s$ under each of those policies. With discount factor $\gamma$ and reward $r(s', z)$, we can define the temporal difference error of a value function for this policy:

$$L_v = E_{(s,s',z)\sim(1-\alpha)D_E+\alpha D_B}(v(s) - (r(s', z) + \gamma v(s')))^2. \tag{2}$$

Leaving aside – for a moment – how the reward can be obtained, we can then utilize this learned value function to find an improved policy by (exponentiated) advantage weighted regression (Peng et al., 2019; Wang et al., 2018) which amounts to weighted supervised learning on the *background* data (which enforces closeness to the policy that generated the data via a temperature $\lambda$) and yields the following policy loss:

$$L_\pi = -E_{(s,s',a)\in D_B} \exp((\gamma v(s') + r(s', z) - v(s))/\lambda) \log(\pi(a|s)). \tag{3}$$

An advantage of this presented scheme is its simplicity and use of well established offline RL methods, allowing for an efficient implementation while utilizing insights from many years of RL research such as the use of target networks and how to deal with terminations (see section 4). The full method is described in Algorithm 1 with the key differences to the offline RL setting marked in blue: the reward source and the mixture of *expert* and *background* data.

**Binary demonstration-based rewards (VfO-bin)** In the simplest setting we avoid any additional learning or estimation bias in the reward function by directly assigning a reward of 1 to *expert* transitions and a reward of 0 to *background* transitions in line with what has been proposed by Reddy et al. (2020). While this might seem trivial, it recovers what a perfect discriminator with infinite capacity would output and removes a layer of complexity. It follows the intuition that we may be able to leverage the value function directly for distinguishing good from bad states in the *background* data, rather than learning an intermediary reward function. We can make this notion more precise by realising that when we only provide a reward of 1 for the *expert* transitions and 0 otherwise, the learned value can be interpreted as:

$$v_{\bar{\pi}}(s_t) = p(z_t = E|s_t) + \gamma E_{a\sim\bar{\pi}(\cdot|s_t),s_{t+1}\sim p(\cdot|s_t,a)}v_{\bar{\pi}}(s_{t+1}), \tag{4}$$

$$= E_{(s_{t+1},s_{t+2},...)\sim\bar{\pi}} \sum_{i=0}^{} \gamma^i p(z_{t+i} = E|s_{t+i}) = \sum_{i=0}^{} \gamma^i p(z_{t+i} = E), \tag{5}$$

which is the cumulative discounted likelihood of futures states having been visited by the expert within the mixed dataset when starting in $s_t$. A policy that maximizes this cumulative return thus prioritises visiting expert states.

A different way to look at it, is that states that are contained in both the *expert* and *background* data will receive a positive and a negative reward signals. Given that policy improvement relies on weighted regression on the *background* data this is perfectly fine: actions that lead to states which are closer to the *expert* data will automatically receive a higher weight.

**Discriminator-based rewards (VfO-disc)**    In a second setting, we consider learning a discriminator to represent the rewards (Ho & Ermon, 2016). When learning from observations, these usually learn to distinguish expert from non-expert states (Zolna et al., 2020; Ma et al., 2022) in order to derive a reward for learning a policy. We adopt the objective from ORIL (Zolna et al., 2020) and pre-train the discriminator by minimizing

$$L_d = E_{s \sim D_E}[-\log d(s)] + E_{s \sim D_B}[-\log(1 - d(s))], \tag{6}$$

where $d(s) \in [0, 1]$ is a binary classifier and the objective is akin to training a discriminator in generative adversarial learning (Goodfellow et al., 2014; Ho & Ermon, 2016). The discriminator output directly serves as reward similar to was done in Wulfmeier et al. (2017) using the Wasserstein-1 (or Earth-Mover) distance (Arjovsky et al., 2017).

## 4  EXPERIMENTS

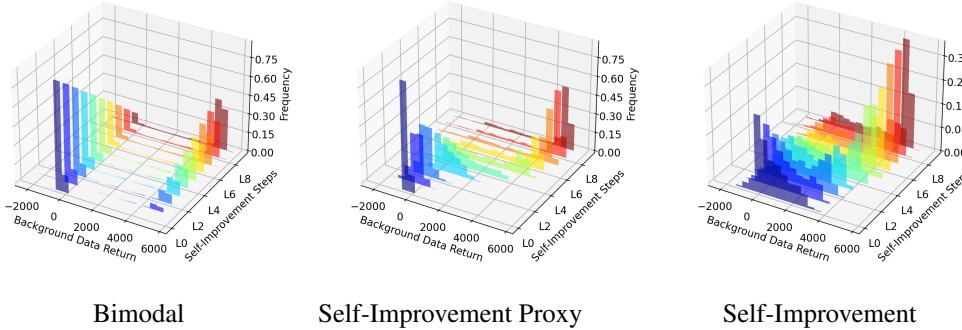

Bimodal                 Self-Improvement Proxy                 Self-Improvement

Figure 1: Schematic visualisation of key differences between bimodal (left) and self-improvement (right) based data configurations. Even this simple visualisation intuits how different algorithmic properties benefit in each setting. Explicitly learning classifiers or discriminators between demonstrations and other data is intuitively easier with stronger split. The intermediate offline self-improvement benchmark serves as a good proxy and enables quick evaluation bypassing serial dependencies between self-improvement steps. The source of all dataset for individual training steps is described in Section 4 in further detail.

We aim to evaluate the two settings schematically shown in Figure 1: bimodal, where the *background* data exhibits a bimodal distribution with a clear gap; and self-improvement, with a more nuanced distribution without a clear expert mode. While the bimodal setting is commonly employed for evaluation in previous work we argue that the self-improvement setting can better serve as an offline proxy for the online iterative self-improvement that motivates our paper. We thus also investigate how representative this proxy is when compared to actual self-improvement style data collection on a subset of the simulation domains. The following questions guide our experiments:

- How do different data distributions affect the performance across algorithms and how effective is VfO? (Sections 4.3 and 4.4)?
- Does the offline self-improvement proxy correlate with full iterative self-improvement (Section 4.6)?
- Can VfO deal with complex inputs such as images (Section 4.5)?

## 4.1 Datasets and Benchmarks

Our experiments are based on two established benchmarks: D4RL (Fu et al., 2021) with many baseline results in the literature and Robomimic (Mandlekar et al., 2021), from which the robosuite (Zhu et al., 2020a) tasks in particular provide a more realistic challenge for our algorithms – and include tasks that require the ability to process image data. D4RL contains data from various OpenAI gym (Brockman et al., 2016) MuJoCo (Todorov et al., 2012) environments, from which we use the Ant, HalfCheetah, Hopper and Walker2D domains. The expert data for these environments comes from policies trained via RL. For each domain, there are 1000 expert demonstrations available; and we drop the action information to obtain the IfO setting. The Robomimic benchmark contains a variety of simulated and real robotics datasets of human demonstrations. From the simulated domains, we use Lift, PickPlaceCan and NutAssembly. For each of these tasks, there are 200 demonstrations.

The most realistic, scalable source of *background* robot data is agent-generated. Therefore in contrast to existing datasets – which are mostly bimodal (very high and low performance) – we target self-improvement as a data source. To emulate the sequential nature of different quality levels that we would expect during self-improvement, we introduce a proxy Self-Improvement Benchmark (SIBench) which uses data generated by a set of varying policies.

To produce data for this benchmark we use a set of policies with various levels of performance; which we train via BC with varying numbers of demonstrations. For each task $\tau$ we train a set of BC policies $\{\pi_\tau^d | d \in \{1, 2, 5, 10, 20, 50, 100, 200, 500, 1000\}, d \leq N_\tau\}$, where $d$ indicates the number of demonstrations that the policy was trained on and $N_\tau$ is the number of demonstrations available. For each of these policies, we collect 1000 episodes, which we use as our *background* data[1]. This setting leads to considerably faster experimental iteration compared to running a full self-improvement experiment (since the data is pre-generated and fixed) and creates a consistent benchmark for fair comparisons; but comes at the cost of removing the data generation or exploration process from the analysis. Importantly, looking at the return distributions for all settings in Figure 1 we can observe that this proposed *background* data is qualitatively close to the data encountered in self-improvement. Further, the different *background* datasets will exhibit different levels of overlap with the *expert* data, ranging from a scenario where the *expert* data is mostly out of distribution to a regime where the *expert* data is contained in the *background* data.

For comparability with prior work and to investigate the added value of our evaluation scheme, we also constructed datasets of what we refer to as bimodal data composed of expert demonstrations and trajectories generated with a random policy (i.e., actions sampled from a uniform distribution). To obtain a more complete picture we sweep over the data mixture: we interpolate linearly from 1000 random demonstrations to 1000 expert demonstration.

Using simulated rollouts of the stochastic policy, we report returns and success rates[2] averaged over 5 seeds and over the last 1e5 training steps to reduce noise (the last 5e5 for Robomimic). We display most results as the difference between the average return observed in the data and the average return obtained from the policy to be evaluated. In contrast to simply reporting policy returns, we argue that this clearly visualises improvement of imitation learning algorithms and allows for better comparison against baselines via improved resolution.

## 4.2 Algorithms and Baselines

We compare our algorithm to a broad set of baselines including behaviour cloning (BC) on the *background* data, BCO (Torabi et al., 2018a), SMODICE (Ma et al., 2022), DILO (Sikchi et al., 2024)[3]. Finally, to provide a performance upper bound and thereby support usefulness of the *background* data, we also report results for Advantage-Weighted Regression (AWR; Peng et al., 2019) trained with ground-truth rewards available for the *background* data. This can be interpreted as an oracle algorithm that does not perform IfO (and is not directly comparable) but serves as indicator of what could be learned from the data. Please refer to Appendix A for further implementation details.

---

[1] The data will be publicly released following the manuscript's review decision.

[2] Note, that this may not be the best metrics for imitation learning performance, as it may provide a distorted view of imitation. E.g. success rate may be blind to any improvement as long as the task is not solved.

[3] Re-implemented in a shared code base for improved comparability. See Appendix B for details. Where possible we verify performance against reported results in prior work.

### 4.3 SIBENCH RESULTS

The results for the D4RL tasks of our SIBench data are shown in Figure 2 (see plots with absolute returns in Appendix E and SQIL with privileged actions in Appendix G). Both VfO-bin and VfO-disc perform well on the Ant and HalfCheetah tasks, getting close to the oracle AWR performance across the full spectrum of *background* data. All methods underperform on the Hopper task. On Walker2D, VfO-disc performs on par with AWR and better than VfO-bin. A possible explanation for VfO-bin's decreased performance could be its lack of immediate reward on the *background* data which could impact its performance on cyclic tasks. SMODICE and DILO perform poorly in comparison on all tasks, only improving on the data on scattered occasions. Generally in this more realistic settings VfO performs remarkably strong; close to 'oracle' performance in many settings despite it having to deal with lack of reward information and lack of action data on the expert demonstrations. The baselines SMODICE and DILO clearly underperform; we hypothesize that similar to residual gradient algorithms in RL (Baird, 1995), which do not make use of stop-gradients or target networks, the signal from the bellman residual may be very weak when there is significant overlap between good and bad trajectories, such as is the case for the *background* data here.

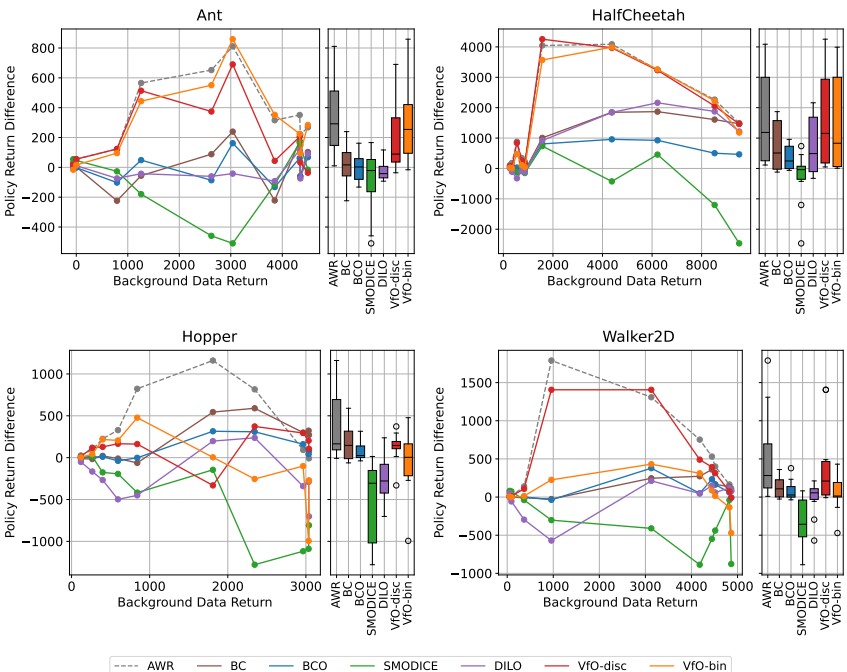

Figure 2: Difference in cumulative return of various algorithms on D4RL tasks using the SIBench data. We plot the average return in the background data against the difference in return (relative to the background data return) achieved by an algorithm. I.e. positive differences mean the policy produced by the algorithm is better than the policy that generated the background data. The boxplots average across the spectrum of background data. AWR, VfO-disc, VfO-bin all show good improvement across the spectrum of *background* data with the oracle AWR performing best. SMODICE and DILO only rarely improve on the data.

Figure 3 depicts the results for the Robomimic tasks of the SIBench data. Overall the results are less conclusive here and this could be related to how improvement is measured: Given that there is no dense reward for these tasks, we resort to success, which is much less indicative of learning progress, i.e. behavior could become more similar to the expert demonstration without higher success rate. Nevertheless, VfO-bin is able to yield positive improvement across all tasks and performs on par or better than AWR, while VfO-disc, SMODICE, and DILO perform worse. Comparing AWR against VfO effectively also compares the underlying driving sources of information, i.e. reward annotations against demonstration. In a scenario with sparse rewards, such as for Robomimic, it is entirely possible that relying on demonstrations allows for better performance.

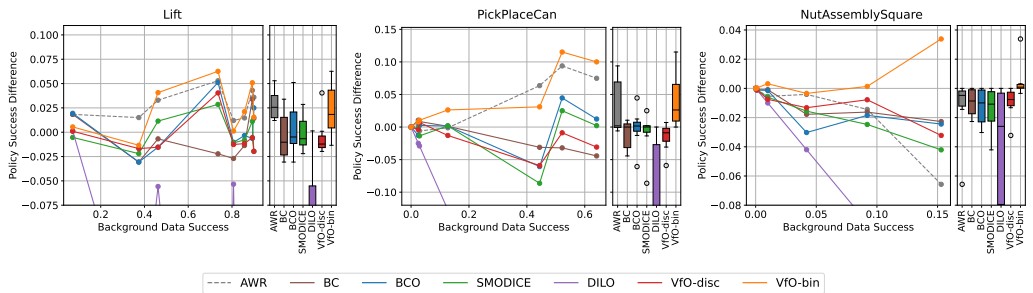

Figure 3: Difference in success of various algorithms on Robomimic tasks using the SIBench data. AWR and VfO-bin mostly yield good improvement. DILO, SMODICE, and VfO-disc have more troubles generating improvement.

## 4.4 BIMODAL RESULTS

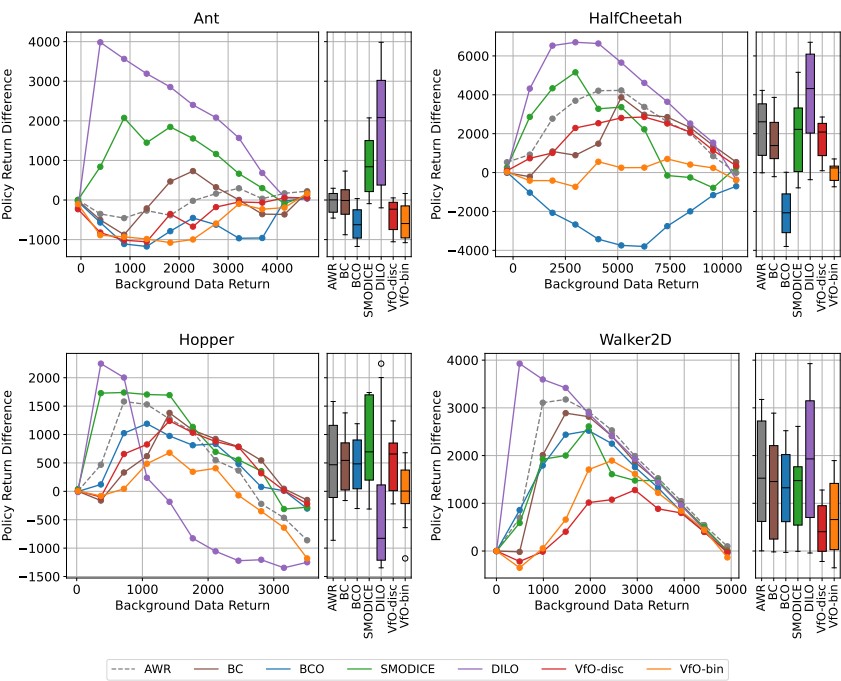

Figure 4: Difference in cumulative return of various algorithms on D4RL tasks using the bimodal data. As reported in previous work, SMODICE and DILO exhibit strong improvement when the data is composed of a little amount of expert demonstrations. VfO-bin and VfO-disc both underperform in that case.

To investigate how SIBench data differs from existing data mixtures and to enable comparability with previous work, we report results for the bimodal D4RL data in Figure 4. DILO and SMODICE exhibit strong improvement, particularly when the data is composed of relatively few expert demonstrations. In most cases, DILO reaches expert performance for the third data point (200 expert demonstrations) which is in line with results from Sikchi et al. (2024). Improvement then degrades smoothly with increasing data quality as there is less room for improvement.

VfO-bin and VfO-disc underperform when compared to BC. The improvement of BC itself can be attributed to the bimodal state distribution and to dynamic effects that can lead BC to pick the more consistent underlying policy (Zhang et al., 2024). These results are also in line with our hypothesis that bimodal data of this type turns imitation learning into a filtering problem of separating the good from bad trajectories and may not effectively measure the properties of imitation learning

algorithms that matter in practical self-improvement scenarios. Instead, the mostly distinct *expert* and *background* state distributions render it more important to pick the right action when the distributions bifurcate. VfO likely struggles to do so because the learned values are not sufficiently discriminative. However, lowering temperatures to increase the effect incurs instabilities.

## 4.5 VISION-BASED RESULTS

We also investigate the ability to learn from image observations (see Figure 5), as is often required in real-world robotics applications. For this we report results on the Robomimic tasks using the SIBench image data. While improvement is more difficult to measure here, we can observe some improvement in the Lift domain for AWR and VfO-bin, again highlighting the fact that our simple VfO scheme is a strong algorithm even in high-dimensional, difficult settings.

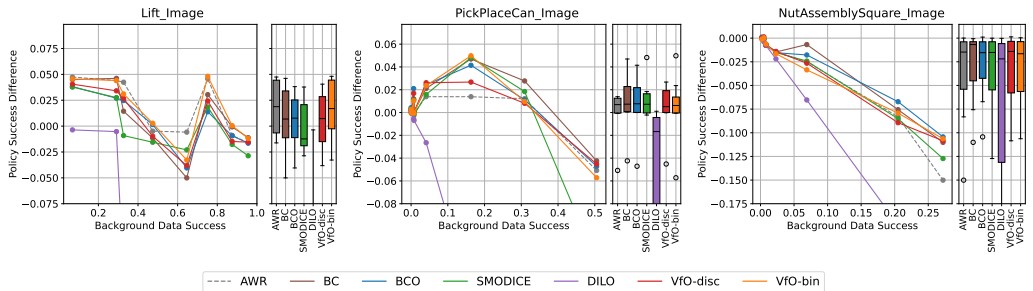

Figure 5: Difference in success of various algorithms on Robomimic tasks using the SIBench image data. In this difficult setup both VfO-bin and AWR manage to achieve some improvement on Lift.

## 4.6 ITERATIVE SELF-IMPROVEMENT RESULTS

To confirm the validity of our SIBench proxy, we run self-improvement experiments akin to what we envisioned in the introduction. In these experiments after learning an initial policy from a seed dataset, we collect 1,000 episodes to form a new dataset for the next learning iteration (and then repeat this process in an improvement loop). We perform 20 iterations in total and seed with data with bad but non-zero performance in order to avoid regions with low signal-to-noise ratios. In order to observe correlation with SIBench we pick VfO-bin as our method to benchmark (for which we expect good performance) and use SMODICE as a baseline. We again run AWR (assuming rewards on all data) to compare to a form of oracle performance. Further baselines are shown in Appendix F.

Figures 6 and 7 plot policy performance against the performance of the input data for each self-improvement iteration. A saw-tooth pattern is observed when consistent improvement is achieved during iterative self-improvement: The achieved performance of one iteration (y-axis) is used as base performance for the next iteration (x-axis), thus the projections onto the diagonal. A box-tooth pattern is observed when performance increases and decreases alternate. This might be caused by oscillating effects, such as the temporary emergence of stationary regions: During one iteration a

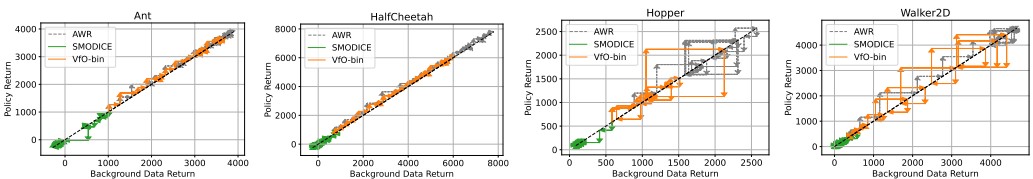

Figure 6: Self-improvement experiments for D4RL tasks. We evaluate VfO-bin, SMODICE, and AWR with ground-truth rewards. Starting with low-performance initial policies, we generate data to train the next iteration of policies for each algorithm and iterate. The policy return (averaged over 1000 episodes) at each iteration is projected on the diagonal and used as *background* data for the next step. Both AWR with ground-truth rewards and VfO-bin lead to strong results.

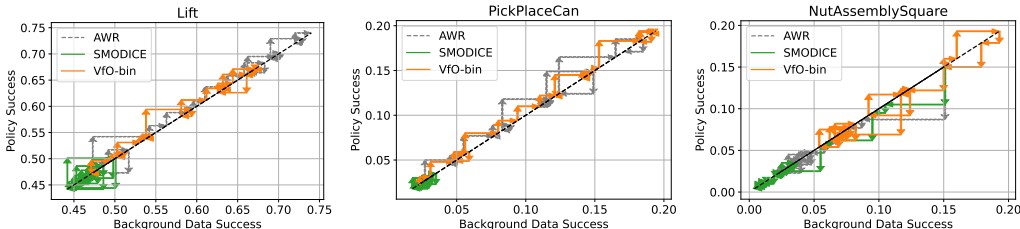

Figure 7: Self-improvement experiments for Robomimic tasks. Starting from different initial performances we can observe whether performance increases or not. Results are mixed, VfO-bin is good on two out of three tasks. Given AWR's dependency on informative rewards (here: sparse), VfO-bin can slightly outperform offline RL in this setting.

policy might decide to remain stationary within a region of high expert density. This would however likely be resolved during the next self-improvement iteration due to an over-proportional visitation of these states.

The plots confirm that whenever we see performance improvement in SIBench we also attain self-improvement when collecting data online, confirming the representative power of our offline proxy. For Ant, HalfCheetah, and Walker2D the self-improvement has not converged within the alotted iterations. Interestingly, for Hopper VfO-bin converges roughly where the zero-crossing of the performance lies in SIBench between 1'500 and 2'000. Additionally, self-improvement is also obtained for Robomimic when starting from data that shows positive SIBench improvement.

Additionally, VfO-bin clearly outperforms SMODICE in this setting and, remarkably, obtains performance similar to the AWR oracle in all settings. We want to highlight that this is a highly non trivial result, bootstrapping imitation learning to mastery via self-collection starting from low signal (near random data) is an open problem in imitation learning (see e.g. Sun et al. (2017)).

## 5    LIMITATIONS AND FUTURE OPPORTUNITIES

While a key application of IfO targets transfer from direct human provided (third person) demonstrations of a task rather than trained first person control, all presented experiments are limited to consistent embodiment between demonstrations and additional data source. Intermediate steps in this direction might utilize state estimation techniques to map correspondences between robot and human states (Luo et al., 2024), but the final goal should remain to exploit existing semantic understanding in pre-trained vision-language and other foundation models (Stone et al., 2023; Yuan et al., 2024; Zitkovich et al., 2023; Wulfmeier et al., 2023; Majumdar et al., 2023). The considerable computational requirements of such models renders the iterative offline learning setting we describe in Section 4.1 more tractable than the pure online learning setting. Self-generated data remains however the most targeted path to obtain the most relevant *background* data for agent training.

## 6    CONCLUSIONS

Imitation learning from observation has the potential to become a principal component of large-scale behaviour learning. We advance this paradigm by suggesting the use of IfO in conjunction with self-improvement. We provide a novel offline benchmark, which we find to be much more representative of this self-improvement setting when compared to existing benchmarks. We also present a simple algorithm (VfO) that builds on ideas from SQIL, ORIL, and AWR to effectively train agents by relying on offline reinforcement learning as the mechanism to learn to imitate. Remarkably, across nearly all experiments in our analysis, VfO is competitive with RL from ground-truth rewards when using just a few action-free trajectories as the defining good behaviour. VfO, and IfO in general, provides an efficient path to real-world RL, where reward function design often becomes the restricting factor.

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

## A    Implementation Details

The policies are simple MLPs with two hidden layers of either 512 units for D4RL or 1024 units for Robomimic. For the VfO algorithms we set the temperature parameter $\lambda$ to 1 on the D4RL tasks and to 0.1 on the Robomimic tasks (see Appendix D for an overview of parameters and a sensitivity analysis). For the methods that learn value functions, we use a target network that gets updated every 200 steps. Some additional implementation details are worth pointing out: We use a multi-scale encoder similar to the one used in the Perceiver Actor-Critic model (Springenberg et al., 2024) that circumvents issues with saturation of non-linearities or insensitivity to lower amplitude signals. This setting reduces dependence on exact input normalisation and can simplify later extensions to multi-task scenarios. Instead of continuous action predictions, we discretize the actions in 101 uniformly spaced bins for which we learn a categorical distribution as is common for recent transformer architectures in control domains (Reed et al., 2022). However, we apply a Gaussian kernel on the last layer to provide sufficient inductive bias in the low data regime. For tasks with terminations, we further bootstrap the values by assuming the agent to continue receiving the same reward, i.e. $v(s_t) = \frac{r_t}{1-\gamma}$. This increases the effect of terminating states and improved performance on terminating tasks. All algorithms are trained for 1e6 learning steps.

## B    Baselines Details

**BCO**    For the BCO (Torabi et al., 2018a) baseline, we train an inverse dynamics model to predict $p(a_t | s_t, s_{t+1})$ on the background data and subsequently use this to label the demonstrations. After this, regular BC learning is done on a $50/50$ mixture of the background data and the now action annotated demonstrations. To implement this, we used the same architecture for the dynamics model as we used for the policies in our experiments.

**SMODICE**    Like in the paper that proposes SMODICE (Ma et al., 2022), we first train a discriminator network to distinguish between the expert and background data based on the observations/states. We apply early stopping at 10000 steps as we found this to be beneficial in preliminary experiments. Subsequently, we learn the value function as suggested and derive weights for weighted BC. Unlike the original paper, we don't apply entropy regularization to policy and don't apply gradient penalization to the discriminator. The SMODICE discriminator has the same architecture as the discriminator for VfO-disc.

**DILO**    This algorithm learns a state-state value function that takes as input *two* adjacent states and does policy improvement via AWR. We implemented the same loss as in Sikchi et al. (2024). Unlike in that work, we found the orthogonal gradient method from Mao et al. (2024) to lead to worse learning stability than simply using the *true-gradient* update in which no target networks or stop-gradients are used. Despite not using the orthogonal gradient update, DILO was still the strongest baseline for the bimodal data. We hypothesize that this difference in results could be due to architectural differences like the discretized actions and multi-scale encoder that we used. For the AWR part of the algorithm we use a temperature of $10$ – note that this would be a setting of $0.1$ in the notation of Sikchi et al. (2024) where the parameter $\tau$ is the inverse of our temperature parameter $\lambda$. We found that increasing the temperature further led to more stable results on the Robomimic but at the cost of essentially turning the algorithm into BC. Otherwise, the settings for this baseline are the same as for the other methods. At the time of writing, the code for DILO is not available yet, so there could be more subtle implementation differences.

## C    Extended Related Work

In addition to the areas described in the main paper, our work strongly relates to and directly builds on research on self-improvement including fundamental reinforcement learning research. The classical idea to have a policy generate its own data to learn and adapt has a long-standing history in RL (Wulfmeier et al., 2023). More recently various works explicitly split the data generation and learning processes (Riedmiller et al., 2022) via model-based (Matsushima et al., 2020) and model-free RL approaches (Lampe et al., 2023; Bousmalis et al., 2024; Springenberg et al., 2024). While many of these works rely on externally defined reward functions, related signals, or vision-language

models as reward sources (Ma et al., 2024), ours directly uses demonstration data to define optimal behaviour (Abbeel & Ng, 2004). Self-improvement research has further gained strong relevance for other foundation model applications such as language modelling (Huang et al., 2022; Choi et al., 2024).

## D  HYPERPARAMETERS

Table 1 provides an overview of the employed parameters. Further Figures 8 and 9 provide hyperparameter ablation for the temperature $\lambda$ and the mixing parameter $\alpha$. They confirm that for both there is a wide range of parameters that enable improvement.

| Hyperparameter | Value |
|---|---|
| learning rate | 3e-4 |
| batch size | 256 |
| MLP layers D4RL | (512, 512) |
| MLP layers Robomimic | (1024, 1024) |
| target network update period | 200 |
| weight decay D4RL | 0 |
| weight decay Robomimic | 0.1 |
| $\lambda$ (temperature) D4RL | 1.0 |
| $\lambda$ (temperature) Robomimic | 0.1 |
| $\alpha$ (mixture ratio) | 0.5 |
| $\gamma$ (discount ratio) | 0.99 |

Table 1: Shared Algorithmic Hyperparameters

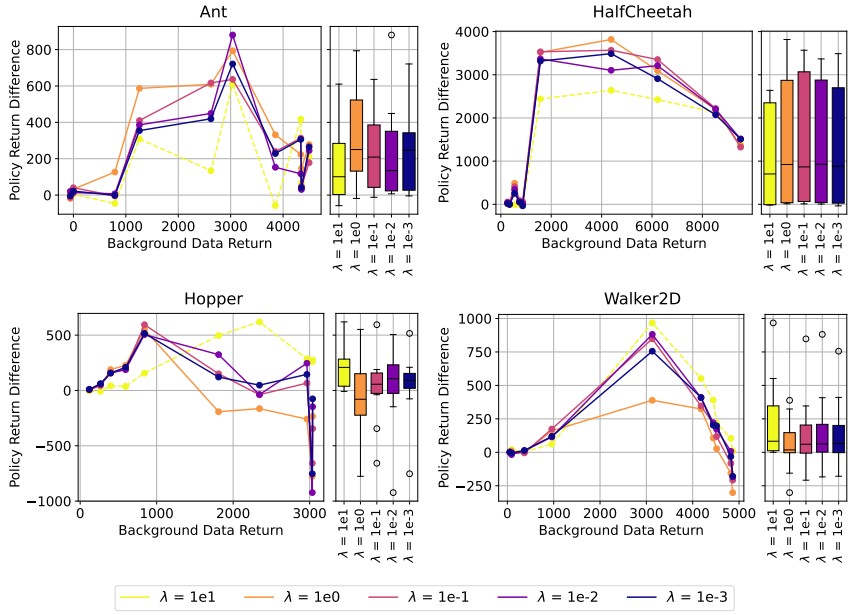

Figure 8: Difference in cumulative return of VfO-bin on D4RL tasks using the SIBench data for different temperature parameters and mixing parameter $0.5$. We plot the average return in the background data against the return of the trained policy. We can observe a fairly wide range of hyperparameter settings leading to improvement.

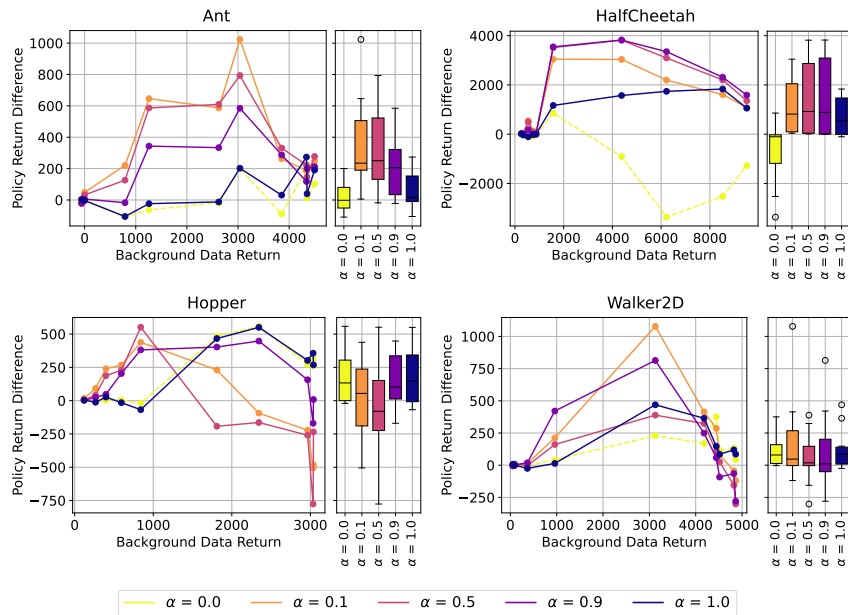

Figure 9: Difference in cumulative return of VfO-bin on D4RL tasks using the SIBench data for different mixing parameters and temperature 1.0. We plot the average return in the background data against the return of the trained policy. Except for Hopper, we can observe that picking a parameter between 0.1 and 0.9 yields consistent improvement.

## E ABSOLUTE RETURN PLOTS

The relative plots in Figures 2 to 5 are not very common. We thus provide the absolute counterparts in Figures 10 to 13.

## F FURTHER SELF-IMPROVEMENT BASELINES

We ran further self-improvement baselines including DILO, BC, and BCO. Figure 14 shows the average returns against the self-improvement iteration. It further confirms that methods bad performance on SIBench are less suited to attain positive self-improvement.

## G OFFLINE SQIL WITH PRIVILEGED EXPERT ACTIONS

Figures 15 and 16 show results for our offline implementation of SQIL with privileged access to expert actions. We provide hyperparameter ablation for the temperature $\lambda$ and the mixing parameter $\alpha$ and can observe that a higher mixing parameter is required to avoid overfitting on the scarce expert actions.

## H IMPROVEMENT PLOTS WITH STANDARD DEVIATIONS

Figure 17 shows the mean and standard deviation of differences in cumulative return of AWR and VfO-disc on D4RL tasks using the SIBench data. Except for Hopper, VfO-disc achieves consistent improvement and exhibits a variance similar to that of AWR.

We further also provide results with an updated evaluation procedure. Here, only the final model weights from each training run are evaluated, using 1000 simulated rollouts each. We keep training with 5 different seeds to capture the distribution of average returns and report mean and standard deviation across training seeds. These are plotted in Figure 18. Adding more rollouts reduces the

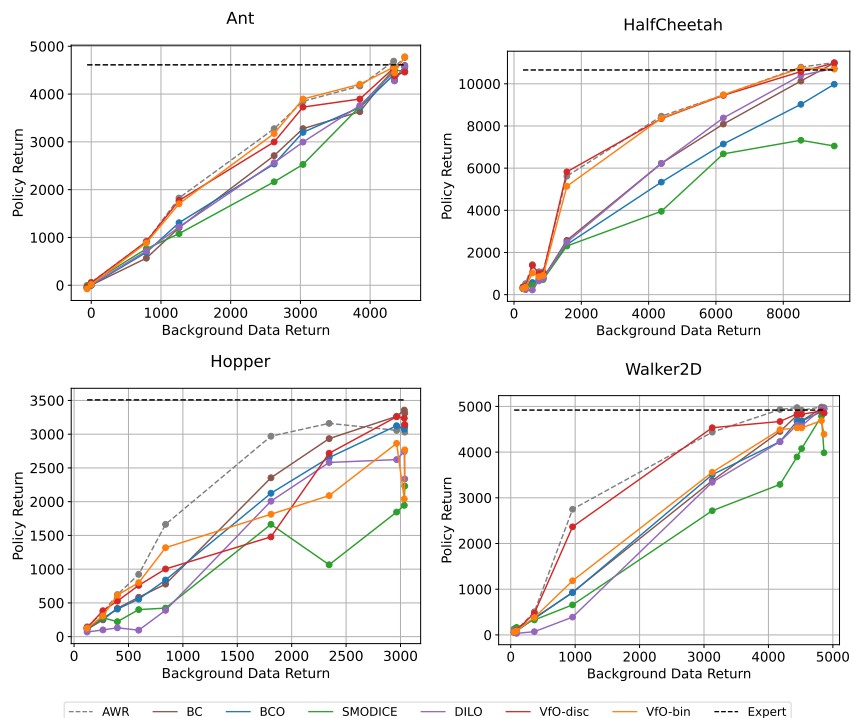

Figure 10: Cumulative return of various algorithms on D4RL tasks using the SIBench data. We plot the average return in the background data against the return of the trained policy. AWR, VfO-disc, VfO-bin all show good improvement across the spectrum of *background* data with the oracle AWR performing best. The corresponding relative plots can be seen in Figure 2.

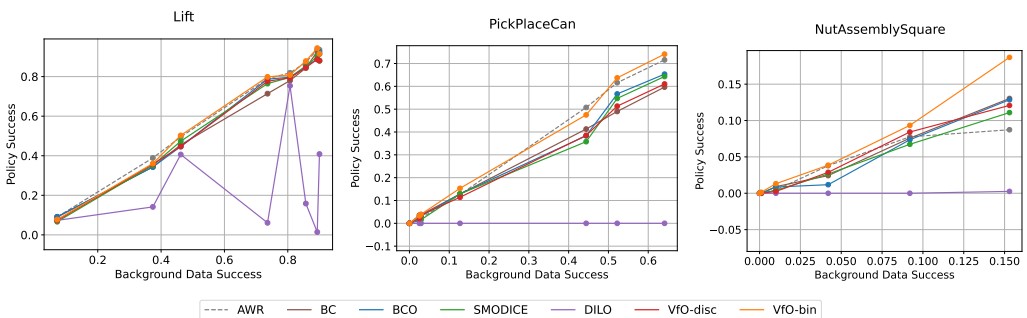

Figure 11: Success of various algorithms on Robomimic tasks using the SIBench data. As the absolute performance range is considerably larger than the differences due to highly different initial data quality, relative rankings require a closer look. AWR and VfO-bin mostly yield good improvement. The corresponding relative plots can be seen in Figure 3.

variance of the results and we can now claim that most VfO model weights attain good performance improvement.

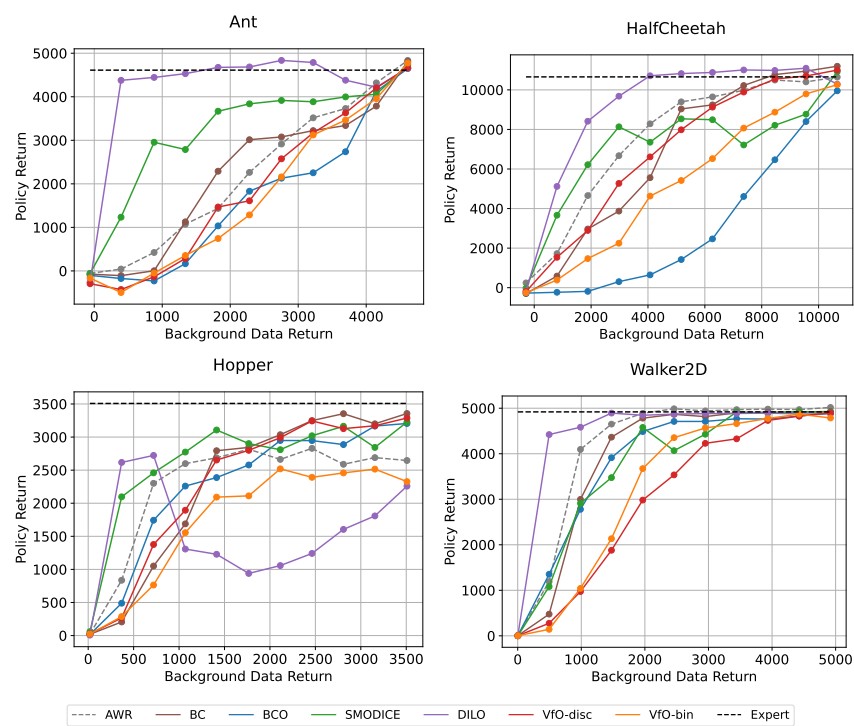

Figure 12: Cumulative return of various algorithms on D4RL tasks using the bimodal data. As reported in previous work, SMODICE and DILO exhibit strong improvement when the data is composed of a little amount of expert demonstrations. The corresponding relative plots can be seen in Figure 4.

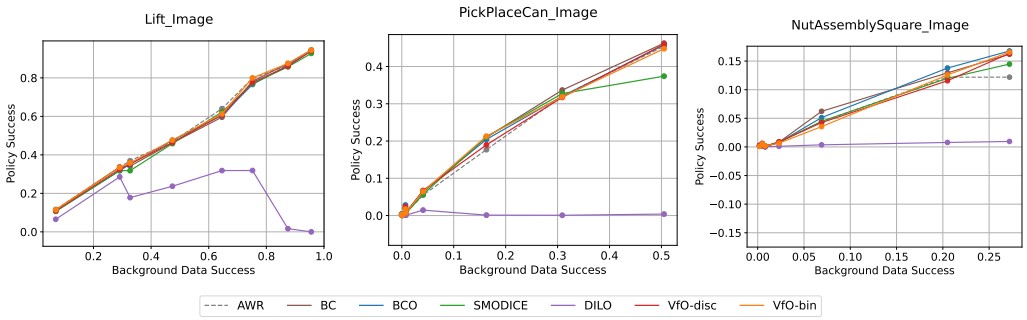

Figure 13: Success of various algorithms on Robomimic tasks using the SIBench image data. Improvement is difficult to discern in these plots. The corresponding relative plots can be seen in Figure 5.

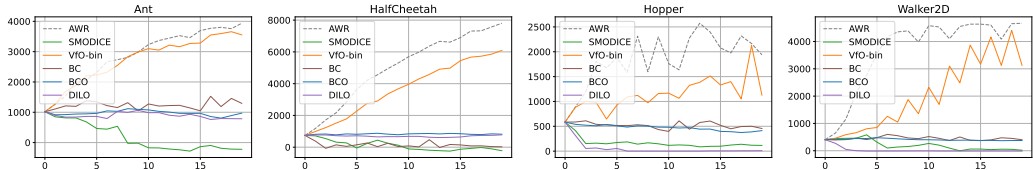

Figure 14: Self-improvement experiments for D4RL tasks. We evaluate VfO-bin, SMODICE, DILO, BC, BCO, and AWR with ground-truth rewards. Starting with low-performance initial policies, we generate data to train the next iteration of policies for each algorithm and iterate. Both AWR with ground-truth rewards and VfO-bin lead to strong results.

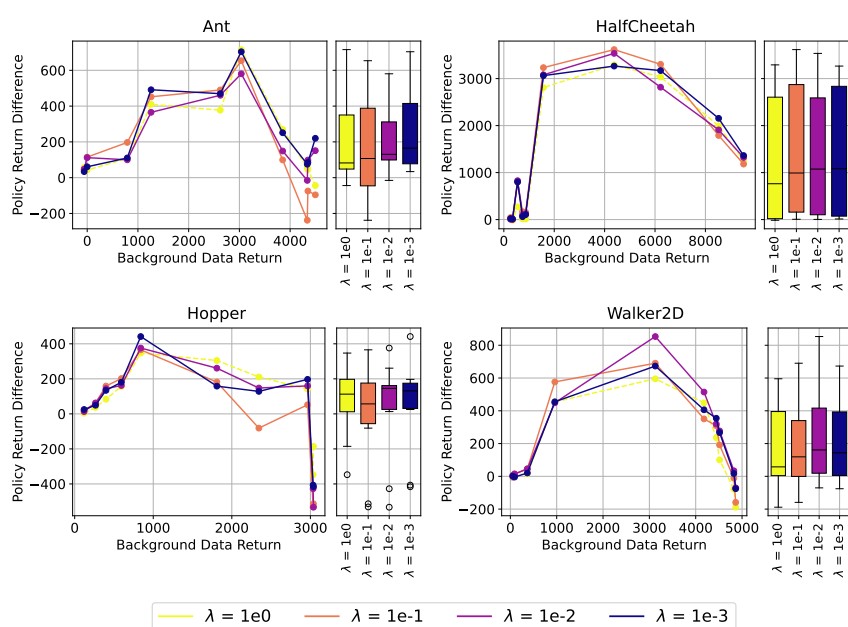

Figure 15: Difference in cumulative return of offline SQIL (with privileged expert actions) on D4RL tasks using the SIBench data for different temperature parameters and mixing parameter $0.9$. We plot the average return in the background data against the return of the trained policy. We can observe a fairly wide range of hyperparameter settings leading to improvement.

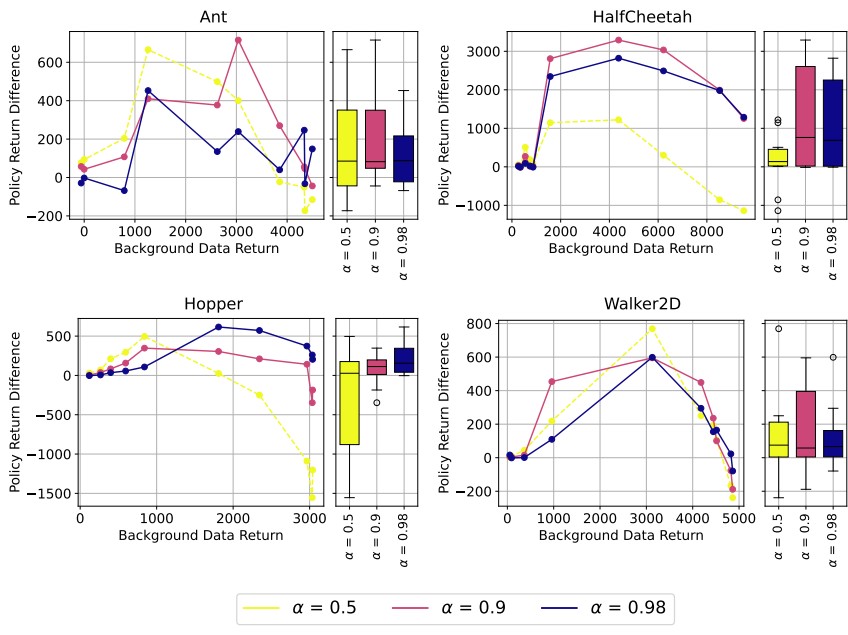

Figure 16: Difference in cumulative return of offline SQIL (with privileged expert actions) on D4RL tasks using the SIBench data for different mixing parameters and temperature $1.0$. We plot the average return in the background data against the return of the trained policy. In comparison to the action-free VfO, we need a higher mixing parameter to avoid overfitting on the scarce expert data.

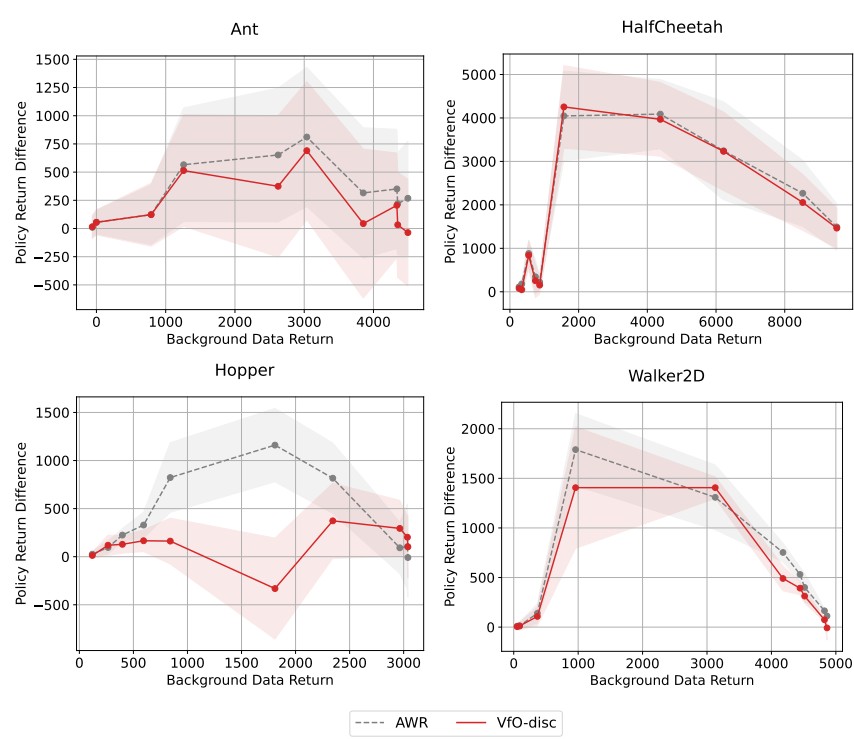

Figure 17: Mean and standard deviation of differences in cumulative return of AWR and VfO-disc on D4RL tasks using the SIBench data. We plot the average return across 5 seeds and across the last 1e5 steps of training.

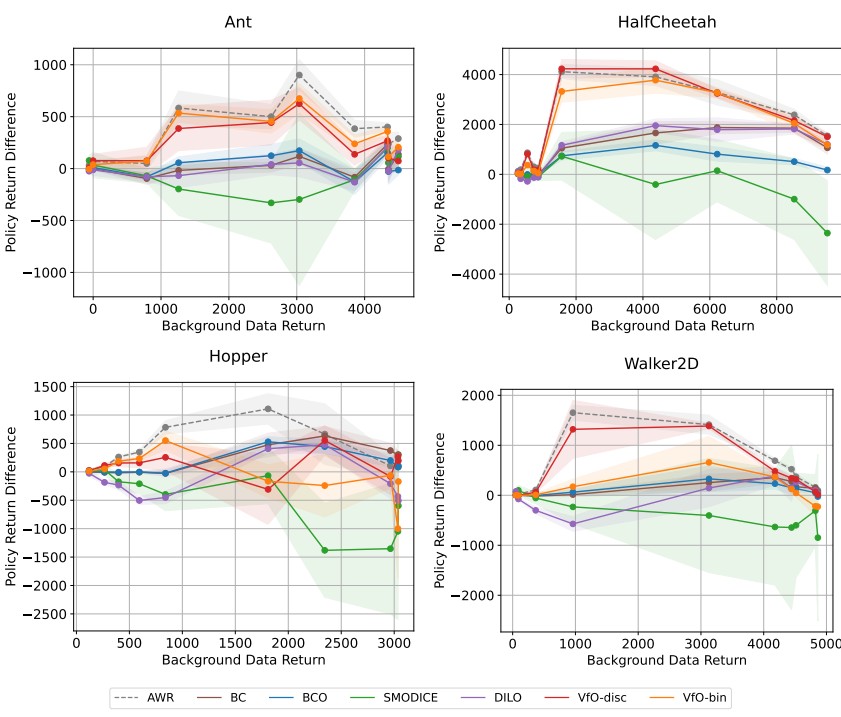

Figure 18: Mean and standard deviation of differences in cumulative return of for different algorithm on D4RL tasks using the SIBench data.

