# OpenReview forum: "Value from Observations: Towards Large-Scale Imitation Learning via Self-Improvement"
_ICLR.cc/2025/Conference — Submitted to ICLR 2025_

### Official Review · Reviewer_5JyX · 2024-10-23

**Soundness:** 3
**Presentation:** 3
**Contribution:** 3
**Rating:** 8
**Confidence:** 3

**Summary:**

The paper suggests a novel method for imitation learning from observation (IfO), a setting where only demonstrations without reward and action labels are available. However, more non-expert experience with action labels may be obtained. The proposed algorithm Value learning from Observations (VfO), extracts information from the expert demonstrations by learning a value function using one of two simple rewards: (i) assign a reward of 1 to expert transitions and 0 to all others (VfO-bin), or (ii) learn a discriminator distinguish expert states and use its output as a reward (VfO-disc). This value function is then used with the offline RL algorithm Advantage Weighted Regression to train a policy. Both versions of VfO are evaluated on datasets obtained from a range of simulated continuous control tasks both in a fully offline and a self-improvement setting. Additionally, benchmark datasets (SIBench) mimicking iterative self improvement are proposed to simplify the evaluation of VfO methods. The experimental results indicate that VfO can improve on the behavior policy used to collect the labeled data. In particular, it can outperform strong baselines like SMODICE and DILO on the SIBench data, sometimes close to AWR with access to the ground-truth reward. In contrast to this, VfO is less successful on bimodal data.

**Strengths:**

The IfO setting (also with iterative self improvement) is highly relevant for scaling up imitation learning, in particular in robotics. The proposed algorithm VfO is quite simple, and easy to implement. An analysis of the VfO-bin variant furthermore shows that the objective tries to maximize visitation of states that occur in the expert demonstrations, which helps with building intuition.

Apart from the VfO algorithm, the paper also proposes a suite of datasets (SIBench) for benchmarking IfO algorithm in a purely offline fashion. Results on SIBench are shown to correlate with the iterative self-improvement setting.

The fact that VfO outperforms more complex baselines like SMODICE on the SIBench data is quite remarkable. Yet, the paper is quite honest in presenting the experimental results, and clearly states that on bimodal data, VfO performs less well.

Overall, the paper is well written and the figures do a good job in conveying the results.

**Weaknesses:**

While the text mentions that the learned value function is essentially the discounted probability of being in an expert state, there is no discussion about why (or when) maximizing this is sufficient as an imitation learning objective. During iterative self improvement, could it not happen that the agent learns to stay stationary in a region of the state space which was visited by the expert even though the expert was not stationary and covered a bigger region of the state-action space? This problem seems to be more severe than in SQIL as SQIL learns a Q-function from expert demonstrations with actions and therefore would not learn to be stationary. I would encourage the authors to discuss the implications of the VfO objective and explain when it can be expected to work well.

In line 325 the choice of the hyperparameter lambda is mentioned (which is different on D4RL and Robomimic). As lambda controls the amount of behavioral cloning, it would be interesting to see its impact on performance, and discuss its role in VfO. I would appreciate a hyperparameter study for lambda.

The labels in figures are too small. This is particularly evident in figure 1 but also true for the other figures to a lesser extent. It would be better to adjust the plots to look good when printed out. I would furthermore encourage the authors to provide additional plots with the absolute performance of the algorithms as it would make it easier to judge how significant the improvements are.

**Questions:**

* In line 334, the bootstrap value is introduced a bit ad hoc. Why is this construction necessary as opposed to not bootstrapping when terminating?
* It would be interesting to discuss if there are qualitative differences between the behaviors learned by the VfO and baseline agents. If such differences exist, can they be related to the training objective?
* It is not clear to me what exactly the subplots with the confidence intervals show (in figures 2, 3, 4, 5). It would be great to explain this in the caption.
* ‘demonstration’ in line 085 does not seem to fit into the sentence.
* The last sentence of the abstract seems to be broken/unfinished.
* ‘VfO-dist’ seems to be a typo in line 341.
* The order of the data points (to which iteration they belong) is a bit hard to make out in figure 6. Maybe encoding the iteration with saturation or some other property could help.

---

> ### Author Response · Authors · 2024-11-20
>
> Thank you very much for your positive, constructive feedback. We particularly appreciate your points regarding the discussion of guarantees and hyperparameter study. All specific responses are added below. Please do not hesitate to reach out for further open questions or comments.
>
> **Improvement Guarantees**
>
> *“there is no discussion about why (or when) maximizing this is sufficient as an imitation learning objective.”*
>
> We worked on different paths towards providing a guarantee for improvement but did not obtain any strict ones. One challenge occurs around the use of a value function representing a different policy than the one that is improved. While some improvement guarantees exist for special cases [1], none matched the present case. Nevertheless, we put rigor into obtaining empirical evidence, a practice that is not uncommon in the field.
>
> [1] Cheng et al., Fast policy learning through imitation and reinforcement.", 2018.
>
> **Stationarity during Self-Improvement**
>
> *“During iterative self improvement, could it not happen that the agent learns to stay stationary in a region of the state space which was visited by the expert even though the expert was not stationary and covered a bigger region of the state-action space?”*
>
> This is an interesting comment. We don’t believe that stationarity during iterative self-improvement should be able to persist: If a policy would be stationary in a certain region this would increase the frequency these states are visited when rolling out the policy. In the next value learning cycle this would then decrease the value of these states which in turn would incentivise the policy to find other states. However, we believe this is a very valuable comment as this could explain the oscillations observed in figure 6 for Hopper and Walker: a policy decides to stay stationary in a region and then manages to resolve it again.
>
> **Hyperparameter Study for Temperature**
>
> *“I would appreciate a hyperparameter study for lambda.”*
>
> We conducted a sensitivity analysis for both the mixing parameter and the temperature and added them to the appendix. These confirm that a mixing parameter of 0.5 (in line with the original SQIL) and a temperature of 1.0 are sensible parameters. However, the analysis also shows that there is a broad range of values for which positive improvement can be achieved.
>
> **Improved Plots**
>
> *“The labels in figures are too small.“, “I would furthermore encourage the authors to provide additional plots with the absolute performance”, “It is not clear to me what exactly the subplots with the confidence intervals show (in figures 2, 3, 4, 5).”, “The order of the data points (to which iteration they belong) is a bit hard to make out in figure 6. “*
>
> Quality of plots is important and we are thankful for the reviewer’s comments. We increased font size for all plots. Provided further information regarding the confidence intervals and encoded iteration progress in Figures 6 and 7. The confidence intervals are taken across the spectrum of background data and summarize the performance of the algorithm.
>
> We believe that the difference based plots make it easier to see whether actual improvement is achieved and improve comparability of methods via better resolution. However, we acknowledge that these types of plot are novel and thus add plots with absolute returns in the appendix.
>
> **Bootstrapping**
>
> *“In line 334, the bootstrap value is introduced a bit ad hoc.”*
>
> The employed bootstrapping generates better results than no bootstrapping. Particularly for terminating tasks (all of the robomimic tasks) this was important as it increases the value on the last states. For cyclic tasks this had no large effect.
>
> **Qualitative Differences**
>
> *“It would be interesting to discuss if there are qualitative differences between the behaviors learned by the VfO and baseline agents.“*
>
> It would indeed be interesting to see if there are qualitative differences between VfO which draws on demonstrations and AWR which draws on rewards, as these are two different sources of information. However during experimentation we did not observe any qualitative differences for the D4RL tasks, maybe because these tasks tend to be solved using very fast motions.
>
> **Typos**
>
> Thank you for mentioning mistakes and inaccuracies, we fixed them.

---

> ### Comment · Reviewer_5JyX · 2024-11-25
> **Response**
>
> Thank you for the additional experiments, improvements to the presentation, and clarifications!
>
> ### Stationarity during Self-improvement
>
> I understand your point about the agent spending more time around a state leading to a lower value estimate. I agree that this should automatically penalize stationary behavior that does not correspond to the expert behavior. You raised an interesting point about the oscillations in Figure 6. Perhaps reducing the amount of new data being collected over iterations could mitigate this.
>
> ### Hyperparameter Study for Temperature
>
> Thank you for adding the sensitivity analysis for lambda and alpha! It indeed looks like the choice 1.0 and 0.5 should provide good results (except for Hopper perhaps).
>
> ### Improved Plots
>
> Thank you for improving the readability of the plots, and providing additional information on the intervals of the box plots. I agree that the plots showing the difference in performance are more interesting. It is nevertheless a good addition to have the absolute values in the appendix.
>
> My other questions and concerns were sufficiently addressed. Thank you!

---

> ### Author Response · Authors · 2024-11-25
>
> Thank you for	your response. We are happy to hear that your concerns were sufficiently addressed and we hope this could improve your confidence in your assessment.
>
> **Stationarity during Self-Improvement**
>
> Your further comment regarding the amount of generated data is interesting and raises the question of how to compose the background data during iterations: Currently 1000 simulated rollouts are produced from the weights of the last self-improvement iteration and used as the sole source of background data for the next iteration (all previous data is discarded). However, experimenting with self-improvement cycles which leverage older data would be very interesting and could also help attenuating oscillations. This may however require an offline RL backbone that better supports off-policy learning (maybe importance weighting could be enough).

---

### Official Review · Reviewer_nrq9 · 2024-10-30

**Soundness:** 3
**Presentation:** 3
**Contribution:** 2
**Rating:** 5
**Confidence:** 4

**Summary:**

The authors propose a simple method for imitation learning with a mixed dataset. The proposed method first labels the dataset with either learned reward or binary reward and then use an offline RL procedure to extract optimal policy. In experiments, the authors compare their method to several offline imitation learning method and achieved superior performance.

**Strengths:**

1. The method is clean and straightforward.
2. The method can outperform selected baselines in both state-based and image-based setups.

**Weaknesses:**

I am concerned about the novelty of the approach:
1. Existing studies (e.g., [1, 2]) already label background datasets using certain imitation learning-based reward functions, then apply offline RL to derive the optimal policy. The proposed method appears to be a variation within this established framework, merely adopting a different implementation. It is unclear what new insights or discoveries are presented here. A more detailed comparison to these works would help clarify the unique contributions.

2. The authors rely on existing reward functions to label the dataset, which further raises questions about the novelty.

References:

[1] Luo et al. Optimal Transport for Offline Imitation Learning. ICLR 2023.

[2] Yu et al. How to Leverage Unlabeled Data in Offline Reinforcement Learning. ICML 2022.

**Questions:**

Suggestions and Questions for Improvement:
1. Consider experimenting with a wider variety of reward functions (as baselines) and discussing best practices for selecting effective reward functions in this context. For example, the use of Optimal Transport (OT) Based Reward used by recent applications [1] and the binary goal completion reward in the goal conditioned setups.

2. Since the author claims that the method targets large-scale imitation learning, it would be valuable to see evaluations on more challenging problems, such as those with longer time horizons or real-world applications. For example, the D3IL benchmark [2].

[1] Haldar et al. Watch and match: Supercharging imitation with regularized optimal transport. In CoRL 2022.
[2] Jia et al. Towards Diverse Behaviors: A Benchmark for Imitation Learning with Human Demonstrations. In ICLR 2024.

---

> ### Author Response · Authors · 2024-11-20
>
> Thank you very much for your knowledgeable feedback. We particularly appreciate the mentioned connection to optimal transport and your point regarding the use of further reward functions and increasingly complex tasks. All specific responses are added below. Please do not hesitate to reach out for further open questions or comments.
>
> **Clarifying Novelty**
>
> *“I am concerned about the novelty of the approach:”*
>
> With respect to the method, the novelty lies in how to use imitation learning-based rewards (such as SQIL or ORIL) in the context of imitation learning from observations (IfO) where no actions are available. We show how to do such an adaptation and how it performs on action-free demonstration, something that has not been done for the mentioned rewards. Further, we sketch a path towards iterative self-improvement, and successfully show how VfO can be applied in such a setting where other algorithms fail. Importantly, we also investigate the lack of representativeness of previous offline benchmarks and propose a novel offline evaluation recipe that is more predictive of iterative self-improvement performance. Our findings are very relevant for the research community as they put previous results under a new perspective and provide improved evaluation and baseline for future research.
>
> **Further Reward Function**
>
> *“Consider experimenting with a wider variety of reward functions (as baselines) “*
>
> We appreciate the reviewers input and the additional references, we will include them in our revision. We also very much agree that further imitation learning-based rewards could be employed. However, we would want to emphasize generality-requirements of the employed reward, e.g. rewards that rely on domain knowledge or rewards that need action annotations would not fit into our paradigm. However, after looking at the paper, the optimal transport based reward would seem to fulfill this requirement and we would be keen to try it out as part of future work.
>
> **More Challenging Problem**
>
> *“it would be valuable to see evaluations on more challenging problems, such as those with longer time horizons or real-world applications.”*
>
> Imitation learning from observations is a young field with many open questions regarding its application and evaluation. We believe that we have made an important contribution by looking into its applicability to self-improvement and providing a more targeted evaluation recipe. By resorting to well-established simulation based tasks we made this paradigm shift tractable for us. While reporting simulation only results is common practice in the field (of all baselines only DILO shows a real-world experiment), we acknowledge the importance of real world experiments and application on more complicated tasks such as the mentioned D3IL benchmark. We will thus focus next on collecting a real world based offline proxy and will also move towards multi-task learning in order to scale data and model sizes.

---

> > ### Author Response · Authors · 2024-11-27
> > **Upcoming Rebuttal Deadline**
> >
> > Thank you again for your helpful feedback. We have improved the paper following the points discussed in our previous message. We further expanded our related work taking into account your suggestions.
> >
> > After positive feedback from all remaining reviewers, we're looking forward to hearing if our improvements address your concerns as well and the rating can be improved. We'd love to clarify all remaining points so please reach out for further questions before the rebuttal end.

---

### Official Review · Reviewer_oqct · 2024-11-02

**Soundness:** 3
**Presentation:** 3
**Contribution:** 3
**Rating:** 6
**Confidence:** 2

**Summary:**

The paper contributes an algorithm for learning from a dataset in which the agent has access to expert demonstrations without action labels, and background datasets with action labels but not actions for the same desired task. In this case, expert refers to being on-task with respect to the desired task for the given embodiment. The paper also contributes a dataset with a variety of policies of different quality on background tasks.

The method uses a value function to transfer information between expert and non-expert data.
Given that the expert dataset doesn’t include actions, the agent must learn the dynamics between actions and states from the background dataset. The state-value function transfers knowledge from the expert data to the background data. The authors note that a state-action value function offline RL approach cannot be applied because we cannot assume any of the background has action annotations that are “good” with respect to the desired task. One variant of VfO assigns binary 0-1 reward to if a state came from background or expert, respectively. Another variant learns a discriminator that performs a soft assignment. Policy evaluation based on the state-value function is performed by computing the loss via temporal difference error of a virtual policy that mixes expert and background data (mixing is controlled by alpha). Then, AWR is used to update the policy. The key different to the offline RL setting is the (soft) binary reward for sourcing a state from the expert versus the background data. The learned policy is incentivized to visit expert states.

**Strengths:**

The results are promising in that they show VfO is competitive with RL from ground truth reward when using a few action free trajectories.

**Weaknesses:**

The assumption that background actions have zero reward may overlook potentially valuable information. For instance, if the background dataset contains partial executions of the desired task (like pouring water for a coffee-making task), discarding these actions might lose important insights that could improve the agent’s performance. I’m curious about the results of a sensitivity analysis of algorithm parameters to better guide practitioners on how they should utilize VfO. For example, how does the ratio of expert to background affect performance, and the alpha parameter? I’m unsure about how compelling the paradigm of VfO is in practice, see question below.

**Questions:**

I’m unsure about how compelling the paradigm of VfO is in practice. Let’s say I have a robot that I’d like to teach a specific desired task: I’d have the capacity to give only a limited amount of expert, on-task demonstrations, but these would contain action labels, and lots of background internet-scale data (likely without action labels, e.g. youtube videos) to train the robot policy. Given that this is leveraging of large-scale datasets is a motivation in the paper, it would be great to discuss why it would be more likely we’d have action-labels for background tasks? It would be helpful to provide concrete examples of real-world scenarios where one would have action-labeled background data but unlabeled expert data, and discuss how common these scenarios are in practice.

How similar does the background data need to be to the desired task? How sufficiently covering of the transitions needed in the expert dataset does the background data need to be? Further analyses to characterize how the similarity between background and expert data affects performance, such as systematically varying the overlap between the datasets, would help readers better understand the generalizability of the approach.

Does the assumption that the background contains actions that should receive a reward of 0 risk losing potentially informative action sequences? It possible that the background data contains trajectories that contain partial executions of desired tasks? For example, if my desired task is preparing coffee, but the background dataset contains pouring water, might we want to learn a positive value for those transitions? It would be helpful to discuss potential ways to extend the method to better handle partially relevant background data, or to analyze how this limitation affects performance in practice.

How does the value of mixing parameter alpha affect the training of the policies? It would be helpful to include an ablation study or sensitivity analysis for key parameters like the expert/background ratio and alpha.

In the bimodal task, why did VfO fail to achieve improvement beyond simple BC? Would an alternative generative policy architecture improve these results? It would be great to further discuss potential hypotheses and suggestions for modifications that might improve its performance on bimodal data.

I am not well-versed in offline RL, and I hope the other reviewers can speak more to the technical appropriateness of the evaluation.

---

> ### Author Response · Authors · 2024-11-20
>
> Thank you very much for your thoughtful feedback. We particularly appreciate your points regarding improved analysis regarding different data configurations and bimodal performance. All specific responses are added below. Please do not hesitate to reach out for further open questions or comments.
>
> **Possibly Good Background Actions**
>
> *“The assumption that background actions have zero reward may overlook potentially valuable information.” “Does the assumption that the background contains actions that should receive a reward of 0 risk losing potentially informative action sequences?”*
>
> Good background data does not represent a problem. On the contrary, with higher returns on the background data the performance of the policy improves. Usually beyond the performance of the background data itself. Also, in the extreme case where the background data exhibits expert performance we do not usually observe degradation (except maybe for Hopper which in our past RL experience can get stuck with suboptimal behaviors). One explanation for this is that our policy only learns from background data. But the value is learned on both expert and background and by assigning a higher reward to the expert data we automatically also increase the value of background states that are similar to expert states. These observations are also in line with the original SQIL paper.
>
> **Sensitivity Analysis**
>
> *“I’m curious about the results of a sensitivity analysis of algorithm parameters to better guide practitioners on how they should utilize VfO.” “How does the value of mixing parameter alpha affect the training of the policies?”*
>
> This is a good comment and we provide a sensitivity analysis for both the mixing parameter and the temperature. These confirm that a mixing parameter of 0.5 (in line with the original SQIL) and a temperature of 1.0 are sensible parameters. However, the analysis also shows that there is a broad range of values for which positive improvement can be achieved.
>
> **Clarification About Expert vs Background Data**
>
> *“I’m unsure about how compelling the paradigm of VfO is in practice.“*
>
> The notion of Expert vs Background data and how we envision these to scale might not have been clearly communicated. We have further clarified both terms in the paper.
>
> Expert data provides the “what”, i.e. indicates what should be done. This is the data we envision to scale to web-scale data. In the present work we use single tasks, but extending this to multiple tasks should be possible without significant adaptation.
>
> Background data: provides the “how”, i.e. an understanding of dynamics (relationship between action and state). This is embodiment specific and we envision it to be collected in a self-improvement cycle to enable an agent to learn any task contained in the expert data.
>
> **Similarity of Expert and Background Data**
>
> *“How similar does the background data need to be to the desired task?”*
>
> The data collected with the different BC policies represents an ablation over background data quality and provides insights into the question of similarity between expert and background data. The background data with low returns has little overlap, the background data with high returns are likely to cover expert data. We can show that across this wide spectrum of background data we are able to consistently attain performance improvements.
>
> **Bad Performance of VfO on Bimodal Data**
>
> *“In the bimodal task, why did VfO fail to achieve improvement beyond simple BC?”*
>
> This is something we were wondering about as well. One possible explanation is the importance of picking the right action in a bifurcation. In the bimodal setting expert and background data are likely to be mostly distinct. It would thus be important to pick the right action once data bifurcates (after which simple BC would be enough). However, VfO seems to struggle doing so. This could be because the weighted policy regression is not strong enough in those bifurcation states which is likely caused by non-discriminative values and high temperature parameters. However, temperature can not be decreased sufficiently without incurring instabilities.
>
> Furthermore, note that experiments are in a low data regime, which is more sensitive to architectural decisions and where inductive biases play an important role. So it is definitely possible that another policy architecture would improve results. But also, these effects may just disappear when increasing the amount of data.

---

> > ### Comment · Reviewer_oqct · 2024-11-27
> >
> > Thank you for your clarifications, I have updated my scores.

---

### Official Review · Reviewer_3CKB · 2024-11-03

**Soundness:** 2
**Presentation:** 3
**Contribution:** 2
**Rating:** 5
**Confidence:** 4

**Summary:**

The paper deals with the problem of learning from a mixture of expert collected data without action annotations(expert data) and another dataset with actions but is not expert collected(background data). The authors pose the Imitation Learning Problem in this setting as an RL problem to learn a Value function over the mixture dataset and then use the value function to learn a policy, adapting two variants of prior methods in RL - SQIL ( VfO - bin) and ORIL (VfO - disc) to compute the pseudo rewards for Value function learning based on the source of the dataset.  The authors also propose a self-improvement benchmark ( SIBench), an offline dataset proxy to online policy improvement compiled from policies learnt at various stages of training starting from a random policy and learnt with behavior cloning.

**Strengths:**

Significance:

Finding edge cases and distribution imbalance in the benchmarks followed in the literature and proving an alternative benchmark. Based on the findings in the paper - the prior benchmarks are biased to be bimodal. Also finding cases where the prior work doesn't perform as well - DILO[1] and SMODICE[2].

Originality:

 It is a mix of ideas from previous work. The algorithm is similar to the one proposed in DILO[1] - learning a value function and using it to learn a policy. Using SQIL to compute the pseudo-reward is new. ORIL[3] style models where a discriminator is learnt to compute the good states has been proposed before in the prior work.

Quality and Clarity:

The presentation is easy to read, the figures are simple but can be improved significantly. There are a few typos which can be addressed in a revised print.



[1] Harshit Sikchi, Caleb Chuck, Amy Zhang, and Scott Niekum. A dual approach to imitation learn- ing from observations with offline datasets, 2024.

[2] Yecheng Ma, Andrew Shen, Dinesh Jayaraman, and Osbert Bastani. Versatile offline imita- tion from observations and examples via regularized state-occupancy matching. In Kamalika Chaudhuri, Stefanie Jegelka, Le Song, Csaba Szepesvari, Gang Niu, and Sivan Sabato (eds.), Proceedings of the 39th International Conference on Machine Learning, volume 162 of Proceedings of Machine Learning Research, pp. 14639–14663. PMLR, 17–23 Jul 2022.

[3] Konrad Zolna, Alexander Novikov, Ksenia Konyushkova, Caglar Gulcehre, Ziyu Wang, Yusuf Ay- tar, Misha Denil, Nando de Freitas, and Scott Reed. Offline learning from demonstrations and unlabeled experience, 2020.

**Weaknesses:**

1. The paper is an empirical experiment on using value functions for a mixture of expertly annotated datasets and background datasets. It would be nice to see rigorous study grounded in theory regarding policy improvement ? What are the bounds of improvement on the policy - the maximum performance that can be achieved by the policy ? Can the policy do better than the expert demonstrations, if yes, in what settings ?

2. There has been a mention of Advantage Weighted Regression(AWR)[1] in Section 3 and AWR is used as the oracle in Section 4, it would be nice to see some equations and proofs on cases where this method meets the performance of Advantage Weighted Regression, instead of just a claim ?

3. Rigorous study and Ablations for the mixture of datasets is missing. Some ideas to explore - What is the maximum performance that can be achieved from just the background data? Methods like learning from Hindsight([2],[3]) can be used to learn without rewards. What is the performance increase with the expert datasets? And if the expert dataset is out of distribution from the background data?

4. Explanations on why SMODICE[4], DILO[5] perform worse ( Other than the overlap of demonstrations) and why the proposed Vfo-bin and Vfo-disc perform better are missing in the experiments analysis ?



[1] Xue Bin Peng, Aviral Kumar, Grace Zhang, and Sergey Levine. Advantage-weighted regression: Simple and scalable off-policy reinforcement learning, 2019.

[2] Andrychowicz,M.,Wolski,F.,Ray,A.,Schneider,J.,Fong,R.,Welinder,P.,McGrew,B.,Tobin,J.,Abbeel, O. P., and Zaremba, W. (2017). Hindsight experience replay. In Advances in neural information processing systems, pages 5048–5058

[3] Ghosh, D., Gupta, A., Reddy, A., Fu, J., Devin, C. M., Eysenbach, B., and Levine, S. (2020). Learning to reach goals via iterated supervised learning. In International Conference on Learning Representations.

[4] Yecheng Ma, Andrew Shen, Dinesh Jayaraman, and Osbert Bastani. Versatile offline imita- tion from observations and examples via regularized state-occupancy matching. In Kamalika Chaudhuri, Stefanie Jegelka, Le Song, Csaba Szepesvari, Gang Niu, and Sivan Sabato (eds.), Proceedings of the 39th International Conference on Machine Learning, volume 162 of Proceedings of Machine Learning Research, pp. 14639–14663. PMLR, 17–23 Jul 2022.

[5] Harshit Sikchi, Caleb Chuck, Amy Zhang, and Scott Niekum. A dual approach to imitation learn- ing from observations with offline datasets, 2024.

**Questions:**

Some questions on the clarity on equations and figures:

In equation 2, it is not clear how do you have access to the expert policy, pi_E(a|s). Is that the learnt policy on expert observations ?

Plots with lines in Figure 6 and 7 are confusing? Could the same idea be conveyed with different style of plots?

Here are some suggestions for stronger results, addressing which I am happy to revise my score:

1. In the algorithm, is it necessary to learn value function and policy in a single iteration ? Since in the binary case, the rewards are always 1 for expert datasets and 0 for the background datasets. Have any experiments been conducted where the value function is learnt just on the expert data and then it is used to learn a policy on background datasets?

2. It could also be possible that if a similar set of states are encountered by the background datasets, it could confuse the value function since all the background states are given a reward of zero ? It would be good to see some results in the extreme case, where all the expert observations are a subset of the background data?

3. Although the Value function is appropriate for a setting without demonstrations, it would also be nice to see how well the algorithm performs when action annotations are available with SQIL. The setting would be SQIL uses the action annotation information, VfO learns just from observations. This would also be a stronger result if VfO can learn comparatively to SQIL like methods without any actions.

4. With reference to the title and the mentions regarding scaling in Section I and - there are already many large pre-trained models available and harder problems like the robotics manipulation problems in the real used in DILO ? It would be nice to see any practical problems being tackled by the proposed methods?(Performance on Carla Leaderboard[1], or using Offline Autonomous Driving Datasets, or learning robotics policies in the real world ) Studies on the number of episodes versus the policy performance and proposing any scaling laws ?


[1] CARLA Leaderboard (https://leaderboard.carla.org/)

---

> ### Author Response · Authors · 2024-11-20
>
> Thank you very much for your in-depth feedback. We particularly appreciate your points regarding lack of theoretical guarantees, additional context regarding data, further experiments, and explanations regarding scaling. All specific responses are added below (split into 3 comments). Please do not hesitate to reach out for further open questions or comments.
>
> **Improved Writing Clarity**
>
> *“There are a few typos which can be addressed in a revised print.”*
>
> We thoroughly proofread the paper, fixed typos, resolved vague statements, reduced redundancy, and improved motivation.
>
> **Lack of Theoretical Guarantees**
>
> *“It would be nice to see rigorous study grounded in theory regarding policy improvement ?”*
>
> We worked on various ideas to provide theoretical guarantees for improvement, however this turned out to be challenging. A main challenge is that we use the value of the mixture policy to improve the target policy. While there are special cases where guarantees can be obtained [1] no such guarantee can directly be applied here. Nevertheless, we put rigor into obtaining empirical evidence, a practice that is not uncommon in the field.
>
> [1] Cheng et al., Fast policy learning through imitation and reinforcement, 2018.
>
> **Surpassing Expert**
>
> *“Can the policy do better than the expert demonstrations, if yes, in what settings ?”*
>
> The primary goal is to imitate the expert as there is no other source of information on what to do. The only effect we intend to leverage in the future is generalization by training on a large dataset of expert demonstrations covering a broad spectrum of tasks. One could consider combinations of imitation and reinforcement learning with external reward signals to explore this direction further but it is out of the scope of the current submission. Further, external reward may not be that amenable to scaling.
>
> **Comparing against AWR**
>
> *“it would be nice to see some equations and proofs on cases where this method meets the performance of Advantage Weighted Regression”*
>
> A theoretical comparison against AWR is difficult. The reason being that both employ a different driving source of information: AWR uses reward annotations (which VfO has no access to) while VfO employs expert demonstrations (which AWR has no access to). So in a sense this also compares said sources of information and scenarios can be constructed where each source is better than the other. For instance in a scenario with dense rewards but scarce expert demonstration AWR is likely to be better. Inversely, a scenario where expert demonstrations are plenty but reward annotations are sparse VfO could be better. This can partly be observed for the robomimic dataset where reward annotations are sparse. It would indeed be interesting to formalize these situations mathematically, but this is a difficult challenge which we leave for future work.
>
> **Data Ablations**
>
> *“Rigorous study and Ablations for the mixture of datasets is missing”*
>
> Studying the property of the data is indeed very relevant. In comparison to other work, we already provide quite a lot of information by clearly taking the quality of the background data into account and by providing the return distribution across the spectrum of background data for an example task. Regarding the maximum performance that could be achieved with the background data, we believe that the return of the data itself, the performance of BC, and the performance of AWR cover this question. Maybe higher performance could be achieved by using privileged information or domain knowledge (e.g. the task being solved in a final state). However, we believe that such information may not be applicable at larger scales.
>
> Regarding the expert being out-of-distribution, examples are provided for this via the lower quality background data, where we can observe that only little improvement is achieved. This however strongly depends on how improvement is measured.
>
> In the extreme case where not even the initial state distributions overlap improvement may not be possible. In such cases exploration would play an important role. However, we envision a scenario where sufficient unlabeled expert demonstrations are available. Partly because we believe that exploration without prior knowledge is ill-posed. Such prior knowledge could originate from data, which brings us back to the scenario at hand.

---

> ### Author Response · Authors · 2024-11-20
>
> **Impacted Performance of DILO/SMODICE**
>
> *“Explanations on why SMODICE[4], DILO[5] perform worse”*
>
> This question has intrigued us for a while. Our current hypothesis is that the Bellman error employed in both algorithms has a weak signal. This is similar to what has been observed in Bellman Residual Gradient algorithms [1]. E.g. for AWR, if we remove target network and stop gradient on the subsequent value, the value learning remains perfectly stable and even reaches a lower TD-error, however the policy derived thereof will have a significantly worse performance. Since SMODICE and our version of DILO also employ this form of free optimization (no alteration of gradients, no target network) we believe that this could be a reason for the reduced performance. Also, note that the published version of DILO employs orthogonal gradient descent which could behave differently, however, the code was not publicly available at submission and our version with unaltered gradients worked better for us.
>
> [1] Baird L., Residual algorithms: Reinforcement learning with function approximation, 1995.
>
> **Single Iteration**
>
> *“In the algorithm, is it necessary to learn value function and policy in a single iteration ? “*
>
> Not sure we understand the comment regarding single iteration? Could you please clarify.
>
> **Expert Policy in Equation 2**
>
> *“In equation 2, it is not clear how do you have access to the expert policy, pi_E(a|s).”*
>
> Equation 2 refers to the virtual policy that the learned value function corresponds to. The algorithm does not need it explicitly, but rather has access to samples thereof. We restructured this paragraph to make this clearer.
>
> **Value Function on Expert Only**
>
> *“Have any experiments been conducted where the value function is learnt just on the expert data and then it is used to learn a policy on background datasets?”*
>
> The value could be trained on the expert data only and this would be equivalent to setting the mixing parameter alpha to 0. However, we found that alpha = 0.5 works well for most cases. We added an ablation of the alpha parameter in the appendix and can observe that training the value on expert data only strongly decreases performance. This is likely due to the fact that we have little expert data and we might observe different results otherwise.
>
> **Expert Subset of Background**
>
> *“It could also be possible that if a similar set of states are encountered by the background datasets, it could confuse the value function since all the background states are given a reward of zero ? “*
>
> A form of this problem is also encountered in SQIL: as the samples (background data) get closer to the expert the effect of the expert decays. However, according to the authors this did not represent a significant problem. Intuitively, if the background data is close to the expert less change to the policy is required. A similar effect is also encountered with GAIL at convergence.
>
> **Comparison Against (Adapted) SQIL**
>
> *“Although the Value function is appropriate for a setting without demonstrations, it would also be nice to see how well the algorithm performs when action annotations are available with SQIL. “*
>
> If we understand right, you are asking for a comparison where privileged expert actions are used. We thus apply the AWR style policy improvement across background and expert action in order to obtain an offline version of the original SQIL paper. We ran these experiments and added them to the appendix. However, given the little amount of expert demonstration we had to increase the mixing parameter alpha to get good performance. Overall performance is similar to VfO, likely because there is not that much expert data.
>
> **About Scaling and Autonomous Driving**
>
> *“With reference to the title and the mentions regarding scaling in Section I” “Studies on the number of episodes versus the policy performance and proposing any scaling laws ?” “(Performance on Carla Leaderboard[1], or using Offline Autonomous Driving Datasets”*
>
> Scaling is an important theme and the present work attempts to lay the foundation for scaling in the future. We envision scaling to take place in the expert data which could incorporate more and more tasks and domains and finally attain web-scale dimensions. Autonomous driving would indeed be an interesting domain: While this usually has action annotations, learning from observations only may be an effective way to use data across different embodiments. However, this is out of scope of the current work and it would require immense additional computational resources. Also, we believe that a natural next step would be to investigate the multi-task setup where indeed scaling laws could be explored since we would move beyond the low data regime.

---

> ### Author Response · Authors · 2024-11-20
>
> **Real World Application**
>
> *“ or learning robotics policies in the real world”*
>
> Real world experiments can be extremely time consuming and given that we are moving in a young field subject to high uncertainty we think that short experimental cycles are of high importance. Thus we mainly focussed on simulated environments, which is also in line with most prior work (from the implemented baselines only DILO shows a real world experiment). However, we agree that frequently testing algorithms in the real world is very important. Given that we have found a good offline proxy for self-improvement, we believe the corresponding data should effectively be collected as a next step.

---

> ### Comment · Reviewer_3CKB · 2024-11-25
> **Clarification on Single Iteration and Summary.**
>
> For each gradient step in the algorithm, is it necessary to update both the value function and the policy simultaneously? One extreme approach would involve fully learning the value function first before optimizing the policy. My question on this topic has been addressed through the discussion on value functions trained exclusively on expert data, which provided some clarity.
>
> I appreciate the authors’ responses to my questions and their effort in incorporating the suggested updates into the manuscript. While I recognize the challenges of conducting real-world experiments and addressing scaling concerns, incorporating such aspects in future work would significantly enhance the practical contributions of this research. Furthermore, deeper exploration of how the concepts are grounded in theory could broaden the work's applicability and impact.
>
> Overall, the revisions and responses have improved the quality of the paper from the initial version. I commend the authors for their efforts, and based on the updates, I am raising my initial score.

---

> ### Author Response · Authors · 2024-11-26
>
> Thank you for your response. We are happy to hear you think the paper has considerably improved. Nevertheless, we would like to better understand why your updated rating remains under the acceptance threshold. Apart from the heuristic approach and missing real-world experiments, is there anything else  that would improve the paper in your opinion? We do believe that the application of IfO to self-improvement as well as the insights into the data dependencies are of very high value for the community.
>
> **Optimization Schedule**
>
> Regarding the training regime, we now understand your original question was about how gradient steps for the value and policy are being coordinated. Currently, it is indeed as you describe, at every training step both value and policy weights are updated concurrently and you are correct that this could be done differently. We initially experimented with a sequential optimization, where first the value would be trained and subsequently the policy. However, this led to slightly worse performance. This is likely due to some optimisation dynamics and is something we want to revisit when we move to the higher data regime.

---

> > ### Comment · Reviewer_3CKB · 2024-12-02
> >
> > Apart from my initial concerns regarding the lack of theoretical connections and the absence of initial results on real-world tasks or experiments at scale, I believe the evaluations and explanations could be improved overall. I appreciate adding the policy return plots; however, the relationships between policy return differences and success differences versus background differences are challenging to interpret, making it difficult to draw clear conclusions.
> >
> > Below are some directions that require further consideration:
> >
> > **Clarification on Plot Details:**
> >
> > - Could you provide additional clarification regarding the plots? Specifically, how are the background return and policy return connected for a given point?
> >
> > - For example, in Figure 12, for a point where the background return on the x-axis is 2000, how is the corresponding policy return computed? Is it derived by summing up per-transition returns for all transitions in the background trajectory? If that's the case, the trajectory rollout for policy return would be different than the background data trajectory.
> >
> > **Scalability and Pipeline Design:**
> >
> > - On the subject of scalability, could you elaborate on the pipeline for handling large-scale data tasks, such as the Open-X Embodiment offline dataset for robotics with over 1M trajectories?
> > Specifically,How should expert data and background data be selected?
> > - Does this method work for practical large scale offline datasets, given that the current evaluation assumes the agent can collect its own data?
> > - What is the recommended ratio of expert data to background data? From the D4RL experiments, it seems the ratio used is 1:10 (e.g., 1,000 expert demonstrations vs. 10 × 1,000 background demonstrations generated by various BC policies). Have you explored alternative ratios, and how does the alpha parameter vary with different data ratios?
> >
> > **Inference of Results:**
> >
> > - The interpretation of the results often lacks sufficient explanation beyond a visual description of the plots.
> >
> > For example, in Figures 6 and 7, while iterative self-improvement works for VfO and AWR, it remains unclear why it does not work for SMODICE. Similarly, in Figure 14, could you provide insights into why other algorithms fail in iterative self-improvement experiments?
> > Was expert data used for the iterative self-improvement case? One hypothesis for Figure 6 is that SMODICE may require expert data, whereas AWR and VfO might perform adequately with background data alone.
> >
> > - Additionally, Figure 14 lacks axis labels, which would improve clarity and understanding.
> >
> > **Analysis of Policy vs. Background Return in BiModal and SIBench**
> >
> > In Figures 10, 11, 12, and 13, which compare policy return and background return on SIBench and Bimodal data:
> >
> > - For the bimodal case, policies tend to achieve higher policy returns even when background returns are low.
> >
> > - In contrast, for SIBench, the slope of the plot generally follows a 45-degree trend.
> >
> > - Notably, for the bimodal case, most algorithms perform better with lower background returns, whereas for SIBench, only VfO-disc (not VfO-bin) achieves higher policy returns in such scenarios. Could you explain this behavior?
> >
> > - Can it be inferred from these results that the bimodal data transfers information more effectively than the SIBench data?

---

> ### Author Response · Authors · 2024-12-02
>
> We very much appreciate your further comments and will provide answers in the following paragraphs (in two OpenReview comments). We have added clarifications regarding these points to the paper as well.
>
> **Clarifications for Difference Plots**
>
> We believe there is a misunderstanding regarding the difference plots. The x-axis represents the average return in the background data. The returns being the sum of ground-truth rewards over all transitions in a trajectory. For every trajectory in the background data we thus sum rewards for all transitions and then average across trajectories.
>
> Once a policy is trained for a specific background dataset, we compute the average return of the policy in an analogous manner but with simulated trajectories obtained with that policy.
>
> Say that for a background dataset with an average return of 2000, we trained a policy that generates an average return of 2500. We plot the difference of 500 against 2000. This has two advantages: we can clearly see whether the policy improves over the background data (if the difference is > 0) and we can report multiple runs across different background datasets in the same plot without losing resolution.
>
> **Clarifications on Background Data for SIBench**
>
> For every SIBench experiment the background data originates from a single BC policy only. The data generated from the different BC policies is not mixed. So for every one of the 10 BC policies that are trained we obtain one corresponding background dataset containing 1000 trajectories. We originally looked into mixing data, but this led to a more complicated protocol. For all D4RL experiments the expert dataset is composed of 10 trajectories and the ratio of expert to background trajectories is thus 1:100.
>
> **Scalability**
>
> Using a concrete example to explain how we envision scaling is a good idea. Lets assume we want to train a policy using the Open-X Embodiment dataset without actions (not relying on action labels enables access to much more data in the long term). We would select all the datasets that have no suboptimal trajectories (which is the majority) and build our expert dataset thereof. Optionally, suboptimal data with action labels could be added to the background data. But if there is no such data, further data would inevitably have to be collected and this could be done using the suggested self-improvement scheme (in the extreme case starting from a random policy). In terms of expert-background ratio, that would bring us into a regime where expert data is abundant (but not very specific). How much background data needs to be collected in this regime is unclear and requires further investigation. As you might be alluding to, this could highlight the importance of being background data efficient.
>
> Regarding the alpha parameter, there may indeed be some dependency on the ratio between the amount of expert and background data. However, as we have shown in our ablation, the range of alpha parameter yielding improvement is fairly wide and we are thus confident that it could be adapted to a different data ratio.
>
> We acknowledge that there are a lot of open questions and variations regarding this approach to scaling in robotics, but so do all other paradigms (BC, prompting, RL). Still, we believe this to be an important direction to explore and we believe the current paper to contribute very valuable initial insights on which future research can be based.
>
> **Bad Performance of Baselines**
>
> All baselines (except AWR) show bad performance on SIBench and self-improvement. For BC we don’t expect improvement as there is no corrective signal. As mentioned in a previous response, for SMODICE and DILO, we believe there is a lack of signal with the Bellman residual like loss if not used in conjunction with stop gradient and target networks (there is a wider design space to explore and these could be adapted but also need further investigation).
>
> For the D4RL self-improvement experiment, access to 10 expert trajectories was provided (VfO would not work without them either). However, the background data is only self-collected and this may indeed be the issue for SMODICE and DILO: there are no (diluted) expert trajectories in the background data.
>
> We agree that the results raise questions. However, we went further than many previous works in our experiments and explored the impact of different types of data sources, including adding self-improvement and a corresponding proxy across a wide spectrum of background data. To a certain extent it is also natural that this raises further questions.

---

> ### Author Response · Authors · 2024-12-02
>
> **Comparing SIBench vs Bimodal results**
>
> Both observations are correct: for Bimodal high returns can be achieved with moderate background data. The reason for this is that a substantial amount of expert trajectories is diluted in the background data. Thus any algorithm that is able to extract them and copy their behaviour will yield high returns.
>
> For SIBench this is more subtle as there may be no trajectories demonstrating expert behavior in the background which makes it much more difficult to obtain large improvements. However, this also makes it more representative of self-improvement. The 45 degree trend mainly indicates little improvement.
>
> Rather than transferring information differently we would claim that Bimodal and SIBench represent different problems. Bimodal is more about filtering out non-expert data and then applying BC. On the other hand,  SIBench is more about locally improving a policy much like RL attempts to do.

---

### Official Review · Reviewer_ghXe · 2024-11-04

**Soundness:** 3
**Presentation:** 3
**Contribution:** 2
**Rating:** 6
**Confidence:** 4

**Summary:**

In this paper, the authors expand on previous soft reinforcement learning methods, to facilitate learning from observations (IfO) without considering actions. In this setting, it is assumed that a reasonable demonstration set exists that does not exhaustively capture the occupancy measure of the target task.

They present their method: Value from Observation (VfO) and evaluate against several baselines on a crafted synthetic dataset, comprised of varying quality rollouts collected from Behavior Cloning policies (BC). The authors’ contribution lies in adapting the SQIL and ORIL in the action-less domain, relaxing the assumptions of the demonstration set needed and showcasing that learning can still be facilitated via an extensive set of experiments.

**Strengths:**

Generally well written paper, with easy to follow structure and well laid out motivations and claims.
The authors make reasonable claims and specifically state their effort to contribute to the significant and challenging problem of learning from Observations, in the offline setting, which can be a prerequisite for large scale learning.

Their experiments are reasonably displayed. The authors compare their method against the baselines in both their own dataset for the popular task in the D4RL and Robomimic benchmarks.

The ablations, especially for figures 6 and 7 are very interesting. They showcase that self-improvent is possible from self collected data, even from a very underperforming starting policy.

**Weaknesses:**

I) Novelty. This author’s contribution lies in showcasing that SQIL type methods can potentially learn even if not considering actions, as long as the demonstration set is reasonably perfomant and in providing a new dataset that could be of use to the community. I believe that this would make a wonderful workshop paper as it further explores the idea that trying to regularize BC that diverges too far from what is demonstrated, can lead to better generalization than simple BC.

II) N

Minor

a) The expert’s returns should be displayed in the plots along the background data. This would give a better understanding to the reader of the impact of the differing quality between the presumed expert data and background data.

b) The choice of  plotting  return differences vs background data returns, can be confusing.

**Questions:**

1) Why does simple BC seem to improve on the background data? Was it not trained to true convergence?

2)What happens when a significant part of the demonstrated data is of very low returns? Is there a point where the method could irrecoverably suffer from trying to imitate these potentially harmful examples?

3) In figures 6 and 7 it can be hard to discern what is happening especially in the later parts of Hopper and Walker. What does this box-tooth behavior mean?

---

> ### Author Response · Authors · 2024-11-20
>
> Thank you very much for your detailed, constructive feedback. We are glad you were able to follow it well and appreciate your points regarding potential for improved clarity on novelty and additional context. All specific responses are added below. Please do not hesitate to reach out for further open questions or comments.
>
> **Clarifying Novelty**
>
> *“Novelty. This author’s contribution lies in showcasing that SQIL type methods can potentially learn even if not considering actions”*
>
> While the method is similar to SQIL, it is applied to a different problem setup for which both an algorithmic adaptation is necessary and for which there are no previous results. Further, we sketch a path for how to learn from observations without access to expert action via self-improvement, and successfully show how VfO can be applied in such a setting where other algorithms fail. Importantly, we also investigate the lack of representativeness of previous offline benchmarks and propose a novel offline evaluation recipe that is more predictive of iterative self-improvement performance. Our findings are very relevant for the research community as they put previous results under a new perspective and provide improved evaluation and baseline for future research.
>
> **Display Absolute Returns Including Expert Return**
>
> *“The choice of plotting return differences vs background data returns, can be confusing.”, “The expert’s returns should be displayed in the plots along the background data.”*
>
> We believe the return differences are extremely valuable in order to clearly see whether the policy is able to improve w.r.t. to the employed background data. Also it improves resolution and allows better comparison of algorithms. Nevertheless, we agree that these plots are uncommon and we thus generate plots with absolute performance, including expert returns, in the appendix. Please note, for the difference based plots expert returns would be on a diagonal which could impact readability.
>
> **Explain BC Performance**
>
> *“Why does simple BC seem to improve on the background data?”*
>
> This is indeed a peculiar behavior. We believe that this is a similar effect to what has been previously reported in generative modes [1]. In heterogeneous data it may be easier for the policy to focus on what is in common or more predictable. If this coincides with better actions then the policy will achieve higher returns. This effect is particularly strong for the bimodal data and will indeed also have a transient effect during training (we train all BC policies for 1e6 steps though).
>
> [1] Zhang et al., Transcendence: Generative models can outperform the experts that train them, 2024.
>
> **Bad Demonstration Data**
>
> *“What happens when a significant part of the demonstrated data is of very low returns?”*
>
> We are not entirely sure what is meant by “significant part of the demonstrated data”. If this refers to background data then such ablation is provided and we observe that we are able to achieve improvement across a wide spectrum of background data quality. If this only refers to the expert data then we do not intend to deal with this case. The goal is to imitate the expert data in all its aspects. The question is rather whether absolute return is a good metric to measure expert imitation.
>
> **Box-Tooth Pattern in Figures 6 and 7**
>
> *“In figures 6 and 7 it can be hard to discern what is happening especially in the later parts of Hopper and Walker. What does this box-tooth behavior mean?”*
>
> The saw-tooth pattern is observed when consistent improvement is achieved during iterative self-improvement: the achieved performance of one iteration (y-axis) is used as base performance for the next iteration (x-axis), thus the projections onto the diagonal. A box-tooth pattern is observed when increase and decrease in performance alternate. This might be caused by oscillating effects. One hypothesis was provided by reviewer 5JyX which hinted that stationary regions might emerge temporarily.

---

> ### Comment · Reviewer_ghXe · 2024-11-26
>
> I carefully went over the revisions and alterations to the paper.
> I also reconsidered the authors response to my initial comment about limited novelty. I believe that there could be value to this direction of self-improving VfO, especially in some applications in robotics, where issues of scale can arise; it is indeed not always possible to capture state-action data. I believe there is merit to this direction, more than a niche case of SQIL.
> I have raised my score to reflect this.
>
> I would like to commend the authors for their improvements to the paper. It reads and motivates the problem better.

---

### Official Review · Reviewer_Qh8y · 2024-11-04

**Soundness:** 3
**Presentation:** 2
**Contribution:** 3
**Rating:** 6
**Confidence:** 2

**Summary:**

This paper proposes a method for learning policies from two types of demonstration data: sub-optimal demonstrations with state-action pairs and expert demonstrations containing only states without action labels. The proposed approach involves learning a value function for the states from both datasets and fitting a policy to maximize this learned value. The value function is supervised in two ways: by assigning a reward of 1 to states in the expert demonstration set and 0 to other states, or by using predictions from a discriminator that determines whether a state is from the expert demonstration set. The method further incorporates iterative self-improvement by generating new sub-optimal demonstrations using the learned policy. The approach is evaluated in various simulation environments from MuJoCo and Robomimic, leveraging open datasets collected in these settings.

**Strengths:**

*Addresses an Important Problem*: The paper tackles the practical challenge of learning from heterogeneous demonstration data, which is valuable in scenarios where expert action labels are unavailable but expert state observations are accessible.

*Novel Methodology*: Extend existing methods that combine value function learning and policy fitting, supervised through either binary rewards or discriminator predictions, which is a creative approach to leveraging available data.

*Application to Diverse Environments*: Applies the method to various simulation environments using open datasets, demonstrating the method's applicability across different tasks and specifically showing good results in self-improvement.

**Weaknesses:**

- *Insufficient Practical Motivation*: The paper lacks clear examples of real-world scenarios where the specific data setting (expert demonstrations without actions and sub-optimal demonstrations with actions) is prevalent.
Providing practical applications, such as the use of shared-embodiment devices like the UMI gripper (https://umi-gripper.github.io/), would strengthen the motivation and highlight the method's relevance to practitioners.
- *Absence of Real-World Experiments*: The lack of real-robot experiments limits the demonstration of the method's effectiveness in practical settings. Including real-world applications would greatly enhance the paper's impact and validate the approach beyond simulated environments.
- *Clarity and Writing Quality*: Several sentences are difficult to read due to approximative language, which affects the overall readability.
For example, the last sentence before the related work section (line 85) and the use of terms like "decade of experience" without scientific backing (line 128) or "struggle to achieve improvement" instead of underperform.
- *Weakness in Related Work Section*: Some citations are not well justified, and the connections to the proposed method are unclear.
This makes it difficult to assess the novelty and positioning of the work within existing literature.
- *Unclear Experimental Results*: It is specified that the reported results are from the training set (line 305). This is concerning, results should be reported from simulated rollouts. The experiments lack clear takeaways, making it challenging to interpret the effectiveness of the method. In Figure 2, the large difference between the discriminator and binary results in the walker experiment is not explained, leaving readers uncertain about the underlying reasons.
- *Lack of Baseline Comparisons in Self-Improvement Experiments*: The second set of experiments demonstrates iterative self-improvement but contains only one relevant baseline and an oracle approach. It makes it difficult to evaluate the advantages of the proposed approach over existing methods.
- *Unsupported Practical Relevance in Conclusion*: The conclusion mentions "practical evaluation settings" and "practically relevant" applications without providing supporting evidence or examples within the paper. These supporting evidence do exist and should be mentionned.

**Questions:**

- *Practical Applications*: Can you provide concrete examples of practical scenarios where expert demonstrations without action labels and sub-optimal demonstrations with actions are available? How would your method be applied in such settings? Is the UMI gripper (https://umi-gripper.github.io/) a good fit?
- *Evaluation Methodology*: How did you evaluate your method? Did you use simulated rollouts? Clarifying this is crucial for assessing the validity of your results. Line 305 mentions success rates, did you mean the policy success difference?
- *Discriminator Performance*: In Figure 2, what accounts for the large difference between the discriminator and binary results in the walker experiment? Does this indicate issues with the discriminator's ability to discern state provenance?
- *Experimental Conclusions*: What are the key takeaways from your experiments? Could you summarize the main findings and how they support the effectiveness of your method? My understanding is that in the simplest settings and data distribution your methods outperform your baselines and approaches the method using oracle reward. While for more complex tasks and data distributions your method underperforms or is similar to the baseline. Is that the conclusion of your first set of experiments?

Additional Feedback:
- *Enhance Motivation with Practical Examples*: Incorporate real-world applications or potential applications where your method would be particularly beneficial. Discuss devices like the UMI gripper or other expert demonstration systems to illustrate practical relevance.
- *Improve Writing and Clarity*:
Revise the manuscript for language and structural clarity.
Ensure that all sentences are clear, concise, and scientifically precise.
- *Strengthen Related Work Discussion*:
Include a section about self-improvement. It seems to be one of the strongest experimental results of the paper but it is not situated in the literature.
- *Clarify Experimental Procedures*: Provide detailed information about the evaluation methodology.
Include an appendix with comprehensive results, such as comparable absolute returns or success rates, to facilitate comparison with other works.
- *Explicitly State Conclusions*:
Draw clear conclusions from your experimental results.
Summarize the main contributions and how they are validated through your experiments.

I would happily raise my score if my concerns about the evaluation methodology are answered.

---

> ### Author Response · Authors · 2024-11-20
>
> Thank you for the very detailed and constructive review. We appreciate your acknowledgement of both the importance of the problem and the novelty of the method. Answers to your comments follow (split in two comments due to length restrictions). Please do not hesitate to reach out for further open questions or comments.
>
> **Practical Motivation**
>
> *“Insufficient Practical Motivation: The paper lacks clear examples of real-world scenarios where the specific data setting (...) is prevalent.”*
>
> We thank the reviewer for pointing out the importance of including examples of real world applications to our work. Shared embodiment devices like UMI are indeed a great point. Using VfO could circumvent the use of a policy interface: Given the demonstrations of the hand-held gripper, VfO could directly be used to collect data on a specific embodiment via self-improvement. Another application would be in settings with motion capture data, where VfO could again be applied to overcome the lack of action annotations. We elaborate on this in the paper and further clarify the long term vision where web-scale data is incorporated as part of the expert data and where an agent collects the background data in self-improvement cycles.
>
> **About Real World Experiments**
>
> *“Absence of Real-World Experiments: The lack of real-robot experiments limits the demonstration of the method's effectiveness in practical settings”*
>
> We agree that frequently testing algorithms in the real world is very important. However, real world experiments can be extremely time consuming and given that we are moving in a young field subject to high uncertainty we think that short experimental cycles are of high importance. Thus we focus on simulated environments, which is also in line with most prior work (from the implemented baselines only DILO shows a real world experiment). Given that we have found a good offline proxy for self-improvement, we believe the corresponding data should effectively be collected as a next step.
>
> **Improved Writing Clarity and Related Work**
>
> *“Clarity and Writing Quality: Several sentences are difficult to read due to approximative language” “Weakness in Related Work Section: Some citations are not well justified, and the connections to the proposed method are unclear. “*
>
> We agree that writing clarity could have been better. We thoroughly proofread the paper, fixed typos, resolved vague statements, reduced redundancy, added context to the related work (including some notes on self-improvement), and improved motivation.
>
> **Clarify Reported Results**
>
> *“Unclear Experimental Results: It is specified that the reported results are from the training set (line 305).”*
>
> The reported results are the returns of simulated rollouts. These are continuously generated during training. In order to reduce noise we average across 5 seeds and across the checkpoints of the last 1e5 training steps. Also, we always use rollouts of the stochastic policy to accurately reflect data collection during self-improvement. We further include plots with absolute returns in the appendix.
>
> **Clear Takeaways**
>
> *“Experimental Conclusions: What are the key takeaways from your experiments?”*
>
> Again, we agree that this could have been clearer. We refined the key questions posed at the beginning of the experimental section, clarified takeaways when discussing results, and summarized them again in the conclusion. There are as follows:
> * There is a high sensitivity of performance w.r.t. the action-labeled background data.
> * VfO performs well on the offline self-improvement benchmark approaching the method using oracle reward. Baselines such as SMODICE and DILO perform well on bimodal background data mixing expert data with very low return data.
> * There is a correlation between performance on the offline self-improvement benchmark and iterative self-improvement experiments, which means that it can effectively be used as proxy for self-improvement.
> * The offline self-improvement benchmark with images is challenging and VfO only achieves partial improvement.
>
> **Explain Performance Difference between VfO-bin and VfO-disc**
>
> *“Discriminator Performance: In Figure 2, what accounts for the large difference between the discriminator and binary results in the walker experiment?”*
>
> One possible explanation lies in how VfO-bin and VfO-disc transfer information from expert to background data: VfO-disc transfers both on a reward and value level, but VfO-bin only uses the value function. During policy learning, VfO-bin has thus no immediate reward. Tasks where this may be important could be impacted. This could explain why VfO-disc is better on cyclic tasks, such as D4RL, where immediate reward could be more important than for non-cyclic tasks, such as robomimic. Why this is particularly large for Walker2D is unclear and requires further investigation.

---

> ### Author Response · Authors · 2024-11-20
>
> **Further Baselines for Self-Improvement**
>
> *“Lack of Baseline Comparisons in Self-Improvement Experiments.”*
>
> Self-improvement experiments are highly costly due to their sequential nature (which is the motivation for an offline self-improvement proxy), explaining the reduced number of experiments. We have however added both DILO and BCO as further baselines to the self-improvement experiments. These confirm the previously observed correlation between offline self-improvement proxy and self-improvement.

---

> ### Comment · Reviewer_Qh8y · 2024-11-23
>
> About the reported results. I am misunderstanding something:
> You should be training on two offline datasets: an expert dataset and a background dataset. My understanding was that you used preexisting expert datasets and pre-generated examples to create the background dataset. In my understanding, training on this data mix does not require rollouts so why are they "continuously generated during training" ?
>
> Regardless, it seems that you are averaging together the results of different models. The model changed because it is being trained between different rollouts.

---

> ### Author Response · Authors · 2024-11-24
>
> Thank you for your further comment. What you describe is correct: training only uses pre-defined expert and background datasets.
>
> In parallel to this an evaluation routine generates rollouts using snapshots of the trained weights at frequent intervals and computes the obtained returns. This generated data is not used for training.
>
> The simulated returns are sensitive w.r.t. to training seed, rollout seed, and when the model weight snapshot is taken. We thus average across all three by using 5 training seeds, generating 10 rollouts per snapshot, and averaging across snapshots of the last 1e5 training steps.
>
> We hope this helps to clarify the experimental evaluation and we will further clarify this in the paper. Please feel free to reach out with any further questions.

---

> > ### Comment · Reviewer_Qh8y · 2024-11-24
> >
> > Thank you for the explanations. Given your answer I understand that you cannot evaluate a single model because results have too much variance over random variations of models. However, for a practical application, a single model has to be deployed and it is the performance of that model that is important. If there is no method to obtain that model and evaluate it, I am worried that your evaluations are not representative of the performance that can be obtained in practice.

---

> ### Author Response · Authors · 2024-11-24
>
> This is a justified comment. However, we want to point out that averaging across training seeds for evaluation is a very established practice. E.g., almost all RL papers report averages across seeds (e.g. [1, 2]) and the same practical difficulty of only being able to deploy a single model arises. Nevertheless, comparison between different algorithms is commonly performed in such a way. Furthermore, averaging across temporal model snapshots once steady state has been attained is very similar to averaging across training seeds.
>
> But in contrast to the established practice we did not report standard deviations. We added plots visualizing the standard deviation in the appendix and plan to fully integrate them as part of the final revision. For simplicity we only plot AWR against VfO-disc and we can observe that a) the obtained standard deviation for VfO-disc is similar to that of AWR and b) most obtained model weights are able to achieve improvement (except for hopper as previously observed).
>
> Finally, we also want to highlight that our self-improvement experiments effectively only use one set of model weights to generate rollouts for the next iteration. And this further confirms that, although subject to stochasticity, improvement can be achieved with the suggested method.
>
> [1] Schulman et al. "Proximal policy optimization algorithms." 2017.
>
> [2] Haarnoja et al. "Soft actor-critic algorithms and applications." 2018.

---

> > ### Comment · Reviewer_Qh8y · 2024-11-25
> >
> > Thank you for the clarification, I am still not convinced that the evaluation methodology is sound, and the fact that other published results use the same methodology is both reassuring about your work and worrisome about the field. Because it seem to lead to good results in your self-improvement experiment, and because you've answered all my questions I will raise my score but I keep a low confidence.

---

> ### Author Response · Authors · 2024-11-25
>
> Thank you for the valuable discussion. We appreciate that evaluation and model selection remain big challenges in behavior learning. This also further highlights the importance of real world experiments where ultimately model weights have to be selected and applied on a real embodiment. We will do our best to collect real world data as part of future work to further advance the practical relevance of imitation learning from observations.

---

> ### Author Response · Authors · 2024-11-26
>
> We have reconsidered your comments and looked into an updated evaluation procedure: Only the final model weights from each training run are evaluated, using 1000 simulated rollouts each. We keep training with 5 different seeds to capture the distribution of average returns and report mean and standard deviation across training seeds.
>
> We generated plots for these new values and added them to appendix H for the time being (Figure 18 is new, Figure 17 is with 10 rollouts over a window of model weights). Compared to the previous evaluation, the variance of the returns decreased, and this is due to generating more rollouts per model weights (the environments exhibit high stochasticity). With this reduced variance we can now claim that most VfO model weights attain good performance improvement.

---

### Author Response · Authors · 2024-11-20

Thank you for the detailed and constructive feedback. We appreciate the predominantly positive sentiment in particular regarding the importance of being able to imitate directly from action-free observations. We also acknowledge commonly emphasized areas for improvements. In addition to individual per-review feedback below, we will summarize main changes here. The paper has been updated with respect to these points including various new experimental results and we believe that this has significantly strengthened its quality and contribution.

**Motivation**

We improved the motivation of the paper by better explaining how we envision the approach to scale to web-scale data: An increasing amount of tasks and domains, including third person demonstrations, are incorporated into the expert data. In case of missing information, data is collected autonomously in self-improvement cycles and integrated as part of the background data.

We further link the suggested approach to more immediate applications such as settings where action labels are difficult to obtain or which would benefit from cross-embodiment transfer, for example autonomous driving datasets [1] or data collected with the UMI-gripper [2].

[1] Carla leaderboard, http://leaderboard.carla.org.
[2] Chi et al.,  Universal manipulation interface:  In-the-wild robot teaching with-out in-the-wild robots, 2024.

**Clarifying Novelty**

While the method is similar to previous work such as SQIL, ORIL, or OTR, its application to action-free demonstrations for which an algorithmic adaptation is necessary is novel. Further, we sketch a path for how to collect missing data autonomously via self-improvement, and successfully show how VfO can be applied in such a setting where other algorithms fail. Importantly, we also investigate the lack of representativeness of previous offline benchmarks and propose a novel offline evaluation recipe that is more predictive of iterative self-improvement performance. Our findings are very relevant for the research community as they put previous results under a new perspective and provide improved evaluation and baselines for future research.

**List of Additional Experiments**

* Hyperparameter sensitivity analysis showing broad region of improvement
* Further baselines for self-improvement confirming representative power of proxy
* Offline SQIL with privileged actions showing performance on par with VfO-bin

**List of Changes**

1. Elaborated on practical applications [Qh8y]
2. Improved writing quality and related work [Qh8y, 3CKB]
3. Clarified reported results [Qh8y]
4. Reported clear takeaways [Qh8y]
5. Explained performance difference between VfO-bin and VfO-disc [Qh8y]
6. Added further baselines for self-improvement experiments [Qh8y]
7. Clarified novelty [ghXe, nrq9]
8. Added absolute return plots in appendix [ghXe]
9. Clarified assumption on expert quality and limited use of prior information [ghXe, 3CKB]
10. Provided further explanation for pattern in Figure 6 and 7 [ghXe]
11. Discussed reward versus demonstrations as source of information [3CKB]
12. Discussed overlap between expert and background data [3CKB, oqct]
13. Explained impacted performance of SMODICE/DILO [3CKB]
14. Added offline SQIL as further baseline with privileged access to expert actions [3CKB]
15. Discussed good background data (SQIL effect) [3CKB, oqct]
16. Clarified how we envision scaling to happen w.r.t. the data [3CKB, nrq9]
17. Provided sensitivity analysis for mixing parameter and temperature [oqct, 5JyX]
18. Clarified role of expert and background data [oqct]
19. Explained bad performance of VfO on bimodal data [oqct]
20. Elaborated on further imitation learning based rewards [nrq9]
21. Discussed risk of stationarity during self-improvement [5JyX]
22. Improved readability of plots [5JyX]
23. Provided further motivation regarding bootstrapping [5JyX]

Please find all further details in the individual rebuttals and please reach out as soon as possible with further comments or questions.

---

> ### Author Response · Authors · 2024-11-25
>
> Thank you again for your valuable feedback. We addressed a majority of the comments thus significantly improving the paper, including a clearer motivation, better writing, and a good batch of further experiments. Given that the end of the rebuttal period is approaching quickly, are there any further questions or comments?

---

> ### Author Response · Authors · 2024-11-30
>
> Dear reviewers, thank you again for your time and effort in reviewing our paper. Your comments were very valuable in improving the presentation of the paper with a clearer motivation, further explanations, and better plots. The experimental section was further strengthened with additional results confirming the applicability of IfO to self-improvement.
>
> We understand that some of you would like to see further theoretical grounding and more real world experiments. However, we believe that their absence is not uncommon in the field. Furthermore, given that we are exploring a direction with high uncertainty, where a majority of baselines can not achieve improvement, it was essential to investigate a setup with fast experimental cycles. This enabled us to generate novel insights around the applicability of IfO to self-improvement and to introduce VfO as an effective baselines, paving the way for future research in the community.
>
> Having said that we would like to ask you if there are any remaining questions or comments we could address during the remainder of the discussion period?

---

### Meta-Review · Area_Chair_biZs · 2024-12-22

**Metareview:**

After careful consideration of the reviews and discussion between authors and reviewers, I recommend rejecting this paper primarily due to concerns about technical novelty and theoretical depth.

The paper proposes Value from Observations (VfO), an approach that adapts SQIL-style methods to work with action-free expert demonstrations alongside action-labeled background data. While the paper addresses a challenge in imitation learning and demonstrates empirical work, the core methodology represents an incremental adaptation of existing techniques rather than a substantial technical innovation.

The main technical contribution lies in extending reinforcement learning-based imitation learning to handle action-free demonstrations through a value function that transfers information between expert and non-expert data. However, multiple reviewers noted that this adaptation, while potentially useful, does not present significant theoretical advances over existing SQIL-type methods. The approach essentially applies known techniques in a slightly different setting without developing new theoretical insights or guarantees.

The empirical evaluation, while thorough within its scope, reveals mixed results. The method shows promise on several benchmarks and can match or exceed baseline performance in certain scenarios. However, it struggles with bimodal data distributions and lacks validation on real-world tasks. The authors' response acknowledging these limitations and proposing them as future work directions is appreciated but does not address the core concerns about novelty.

The introduction of SIBench for evaluating self-improvement capabilities represents a useful contribution to the field. However, this dataset contribution alone is insufficient to overcome the paper's limitations in theoretical advancement and practical demonstration of scalability - a key claim made in the title and motivation.

The authors have been responsive during the discussion period and made earnest efforts to address reviewer concerns through additional experiments and clarifications. However, the fundamental issues regarding novelty, theoretical grounding, and practical applicability persist. Several reviewers, while acknowledging the paper's empirical contributions, expressed that this work might be more suitable as a workshop contribution that explores existing ideas rather than a full conference paper advancing the field in significant new directions.

Given ICLR's emphasis on novel technical contributions and theoretical advances, I recommend rejection while encouraging the authors to further develop the theoretical foundations and practical applications of their approach. This work could make a stronger contribution with more theoretical analysis of when and why the method works, demonstration of practical benefits in challenging real-world scenarios, and clearer differentiation from existing methods.

**Additional Comments On Reviewer Discussion:**

The rebuttal period generated substantial and constructive discussion between the authors and most reviewers, leading to meaningful improvements in the paper's clarity and technical depth. While reviewer Qh8y participated in discussions, the comments demonstrated limited understanding of the field's practices.

The most substantive technical discussions came from reviewers 3CKB and nrq9, who raised important concerns about theoretical foundations and novelty claims. The authors acknowledged the challenges in providing theoretical guarantees but demonstrated through additional experiments that their empirical validation approach aligns with field standards. While this partially addressed the concerns, the lack of theoretical grounding remains a limitation.

Reviewer 5JyX provided technical feedback regarding potential failure modes and stationary behavior during self-improvement. The authors' responses to these points led to insights about oscillation patterns in their results and prompted additional analysis that strengthened the paper.

The practical applicability and scaling claims were examined by reviewers oqct and nrq9. The authors responded by better articulating their vision for scaling through web-scale expert data collection and clarifying the distinct roles of expert and background data. However, the absence of real-world validation or large-scale experiments remains a limitation.

Reviewer ghXe initially questioned the novelty but was ultimately convinced by the authors' clarification of their contributions, particularly regarding the adaptation to action-free demonstrations and insights about self-improvement. This perspective shift was noteworthy but did not fully offset the remaining concerns about incremental advancement raised by other reviewers.

In weighing these points for the final decision, the area chair focused primarily on the substantive technical feedback from reviewers 3CKB, nrq9, and 5JyX regarding theoretical foundations and practical validation. While the authors made commendable efforts to address these concerns through additional experiments and clarifications, the fundamental issues about limited theoretical grounding persist. These limitations, central to ICLR's emphasis on technical innovation, form the basis for the rejection recommendation.

---

### Decision · Program_Chairs · 2025-01-22

Reject